# Executive functions mediate the relationship between cardiorespiratory fitness and academic achievement in Spanish schoolchildren aged 8 to 11 years

María Eugenia Visier-Alfonso[1,☯,¤a], Mairena Sánchez-López[2,3,¤b,‡]*, Vicente Martínez-Vizcaíno[2,4,‡], Estela Jiménez-López[1,2], Andrés Redondo-Tébar[2], Marta Nieto-López[5]

**1** Faculty of Nursing, Universidad de Castilla-La Mancha, Cuenca, Spain, **2** Health and Social Research Center, Universidad de Castilla-La Mancha, Cuenca, Spain, **3** School of Education, Universidad de Castilla-La Mancha, Ciudad Real, Spain, **4** Faculty of Medicine, Universidad Autónoma de Chile, Talca, Chile, **5** Faculty of Medicine, Universidad de Castilla-La Mancha, Albacete, Spain

☯ These authors contributed equally to this work.
¤a Current address: Faculty of Nursing, Universidad de Castilla-La Mancha, Cuenca, Spain
¤b Current address: School of Education, Universidad de Castilla-La Mancha, Ciudad Real, Spain
‡ These authors also contributed equally to this work.
* mairena.sanchez@uclm.es

**Data Availability Statement:** Data are available from figshare: https://figshare.com/s/

## Abstract

### Objectives

Previous research has studied the influence of physical fitness on academic achievement through executive functions. However, the nature of this relationship remains unclear. This study aimed to analyze how cardiorespiratory fitness (CRF) and executive functions are associated with academic achievement and to examine whether the relationship between CRF and academic achievement is mediated by executive functions in schoolchildren.

### Methods

This was a cross-sectional study including 570 schoolchildren, aged 8 to 11 years, from Cuenca, Spain. Data were collected from September to October 2017. Sociodemographic variables, family socioeconomic status, pubertal status, academic achievement, CRF (20-meter shuttle run test) and executive functions (inhibition, cognitive flexibility and working memory, NIH Toolbox battery in Spanish, v 1.8; iPad Pro, Apple, Inc.) were measured.

### Results

Overall, ANCOVA models controlling for age, gender and mother educational level showed higher scores in language and mathematics in children in higher categories of CRF, inhibition, cognitive flexibility and working memory than in children in lower categories. The effect sizes were moderate ($p < 0.05$, partial eta squared: from 0.05 to 0.12). Moreover, the mediation analysis showed that inhibition partially mediated the relationship between CRF and language (c' = 0.058; IC = [0.005; 0.028]) and mathematics (c' = 0.064; IC = [0.005; 0.030]) grades. Similarly, cognitive flexibility mediated CRF's relationship with language (c' = 0.059;

d032c7e016f2bf3dd4b4. Figure: 10.6084/m9.
figshare.9746126.

**Funding:** This study was funded by the Ministry of
Economy and Competitiveness-Carlos III Health
Institute and FEDER funds (FIS PI19/01919) to
VMV. Additional funding was obtained from the
Research Network on Preventative Activities and
Health Promotion (RD12/0005/0009) to VMV. ART
are supported by a grant from the University of
Castilla-La Mancha (2018-CPUCLM-7813). The
study sponsor has no role in study design,
collection, analysis, interpretation of data, the
writing of the report or the decision to submit the
manuscript for publication. http://www.isciii.es/
ISCIII/es/contenidos/fd-investigacion/financiacion.
shtml

**Competing interests:** The authors have declared
that no competing interests exist.

IC = [0.003; 0.028]) and with mathematics (c' = 0.066; IC = [0.003; 0.029]); however, a significant relationship remained. For working memory, mediation analysis showed no significant results (c' = 0.92; IC = [-0.002;0.025] P > 0.05 in language; c' = 0.103; IC = [-0.002;0.029] P > 0.05 in mathematics). Mediation ranged from 13.38% to 36%.

## Conclusions

Children in higher categories of both CRF and executive function showed higher grades in mathematics and language. The findings indicated that a significant proportion of the positive influence of CRF on academic achievement was mediated by improvements in inhibition and cognitive flexibility. Thus, this study supports the hypothesis that improvements in CRF may contribute to increasing academic achievement not only through a direct mechanism but also through improvements in executive functions.

## Introduction

Academic achievement is defined as the extent to which a student has achieved their educational goals [1], and it is commonly measured by grade point average or standardized tests [2]. High academic achievement is considered a key aspect of later development, predicting occupational and social success across the life course, whereas low academic achievement is linked with leaving school and facing health problems [3]. Academic achievement is a multicausal concept determined by personal, social and institutional/school factors [2].

Among the personal factors that influence academic achievement, cognitive processes have been widely studied [4–6]. Cognitive processes are in charge of directing thoughts and actions, formulating decisions and guiding behavior toward specific goals [1]. Executive functions are a set of cognitive operations, three of the most studied are inhibition, working memory and cognitive flexibility [7]. Executive functions are considered necessary for learning and are positively associated with academic achievement in children and adolescents [8].

Physical fitness is positively associated with academic achievement [9,10]. Previous studies have shown that children with higher levels of fitness (particularly cardiorespiratory fitness, CRF) have better academic results than their unfit peers [1,9,11,12]. Similarly, higher levels of CRF are related to a positive effect on cognitive functioning in children [1,13]. However, the degree to which executive functions may modify, mediate, or confound the association between CRF and academic achievement in children is uncertain.

To our knowledge, only three studies have analyzed the mediating effect of executive functioning in the relationship between physical fitness and academic achievement in children [14–16]. These studies have shown significant a positive relationship between physical fitness and academic achievement mediated by executive functions, suggesting that gains in physical fitness can improve cognitive processes. However, two of such studies [14,15] modeled their mediation analyses considering physical fitness as an umbrella term that includes muscular strength and CRF, although the evidence of the relationship between muscular strength and academic achievement is far from consistent [17]. Moreover, no study has reported mediation analyses distinguishing among the different domains of executive function in the association between CRF and academic achievement. Additionally, no study has provided results separately by gender, even though gender differences in the association between CRF and academic

achievement have been reported [17]. Moreover, differential brain maturation and executive function development between boys and girls have been reported [18].

This study aimed to (i) analyze the association between CRF and the executive function domains with academic achievement (language and mathematics) and the association between the three executive function domains and academic achievement, controlling for potential confounders, and (ii) examine whether the association between CRF and academic achievement is mediated by executive functions in schoolchildren.

## Methods

This was a cross-sectional analysis of the baseline data from a cluster-randomized controlled trial (NCT03236337), aimed at assessing the effectiveness of a physical activity program (MOVI-daFit!) to diminish cardiometabolic risk and to improve CRF, executive functioning and academic achievement in schoolchildren. The design, sampling procedures and methods for collecting the variable data have been described elsewhere [19].

### Participants

This study included 570 schoolchildren, aged 8 to 11 years, belonging to 10 schools in Cuenca, Spain. In these schools, all fourth- and fifth-grade children were invited to participate, except for those who met any of the following exclusion criteria: the children had Spanish learning difficulties; teachers or parents reported that the children had serious physical or mental disorders that could impede their participation in the program activities, or pediatricians reported that the children had chronic disorders such as heart disease, diabetes, or asthma that could prevent their participation in the program activities. Finally, our analysis included 563 participants for whom data on all study variables were collected. This study was approved by the Clinical Research Ethics Committee of the Virgen de la Luz Hospital in Cuenca (REG: 2016/PI021) and by the Board of Governors of each school. After obtaining approval, we invited all of the children's parents to a meeting in which we presented the objectives of the project and asked them to sign an informed consent form for their children´s participation. Parents were informed that they could revoke the participation agreement at any time. Every child was verbally asked to provide consent prior to the completion of each test.

The research was carried out by a multidisciplinary team composed of nurses, psychologists, teachers and physical activity experts. Each test of the study was carried out by the most appropriate team members (e.g., anthropometric tests were carried out by nurses, cognitive tests by psychologists, physical activity tests by sports science educators).

### Variables

The measurement procedures used in this study have been extensively described elsewhere [19]. The study variables were measured from September-October 2017 in all schools by trained researchers following standard procedures. Data were collected during two sessions two weeks apart. On the first visit, physical tests were performed, and demographic information -age, gender, pubertal stage and mother educational level- was collected. On the second visit, participants underwent a cognitive evaluation.

Anthropometric measures, weight and height, were measured in standardized conditions. The mean of the two measurements of weight and height was used to calculate the body mass index as the weight in kilograms divided by the square of height in meters (Kg/m2).

Academic achievement was measured with data provided by the schools. These data consisted of children's final grades in language and mathematics from the previous year (2016/17, 3rd and 4th grades). In Spain, academic achievement is measured on a 0 to 10 scale, with 10

being the maximum grade. Final grades represent the student´s work in an entire academic year, as assessed by the teacher in a continuous evaluation process.

CRF was assessed by using the 20-meter shuttle run test, a validated test to estimate aerobic capacity in children [20]. Participants ran 20 meters between two lines while keeping pace with audio signals emitted from a prerecorded CD. The speed was initially 8.5 km/h and then increased by 0.5 km/h for each stage (1 stage equals 1 minute). The last half stage completed by the child was considered an indicator of CRF. Maximal oxygen intake ($VO_{2max}$) was estimated by applying the Leger formula [21].

Three executive functions were assessed by using the NIH Toolbox (NIH Toolbox in Spanish, v. 1.8) [22]. All measurements were performed individually in a quiet room using a tablet (iPad Pro, Apple, Inc.). The assessment tests included in the Toolbox for each cognitive domain are listed as follows.

**Inhibition.** Was assessed using an adaptation [23] of the Flanker test. Children were required to indicate the left-right orientation of a centrally presented stimulus while inhibiting their attention to the potentially incongruent stimuli surrounding it. In some trials, the orientation of the flanking stimuli was congruent with the orientation of the central stimulus ($> > > > >$ or $< < < < <$,) and in other trials, this orientation was incongruent ($> > < > >$ or $< < > < <$). The Toolbox included a practice block of trials, and if passed, a 20-trial block was presented. The trials consisted of a sequence of congruent and incongruent sets of arrows. Using a two-vector method that incorporated both accuracy and reaction time, a final score was calculated for participants who maintained a high level of accuracy ($> 80\%$) as follows: (0.25 x number of correct responses) + 5 –LOG10 [(congruent reaction time + incongruent reaction time / 2)] [23]. For children scoring $< 80\%$, a total score considering accuracy was calculated.

**Cognitive flexibility.** Was assessed using the Dimension Change Card Sort (DCCS) [23]. Children were shown two target cards and asked to sort a series of bivalent test cards, first according to one dimension and then according to the other dimension. After a four-trial practice, children were presented a 30-trial block with both "shape" and "color" requirements. With accuracy percentage and reaction time on preswitch and postswitch, a raw score was obtained [23]. Using a two-vector method that incorporated both accuracy and reaction time, a final score was calculated for participants who maintained a high level of accuracy ($> 80\%$) was as follows: (0.167 x number of correct responses) + 5 –LOG10 [(congruent reaction time + incongruent reaction time / 2)] [23]. For children scoring $< 80\%$, a total score considering accuracy was performed.

**Working memory.** Was assessed with the List Sorting Working Memory Test [24]. Participants were presented a series of pictures, and items were presented both visually and auditorily. Then, participants were required to repeat the names of the items observed in order of size, from smallest to largest. The number of items presented in each trial increased by one each series. This test was composed of two parts. In the first part, all items were from the same category (animals or food items). In the second part, items from both categories (food items and animals) were presented together. In this part, participants had to repeat the items by category and size. Final scores were based upon a sum of the total correct trials across the two lists, which comprised the List Sorting "Total Score" [24].

To asses socioeconomic status, we surveyed parents using the scale proposed by the Spanish Society of Epidemiology [25], with items related to employment and the highest level of education obtained by both the father and mother. Five SES categories were obtained: lower, lower middle, middle, upper middle and upper. Because most authors have reported that mother education level is a strong predictor of children's academic achievement, we obtained this variable using items related to the highest level of education obtained by the mother from this

questionnaire. We thus obtained a five-category index: no reading/no studies, elementary studies, secondary studies, high school and university studies.

## Data analyses

Statistical analyses were performed using IBM SPSS Statistics v.24 0 (IBM Corp., Armonk, NY, USA), and the level of significance was set at $p < 0.05$. The correlation matrix plot was carried out in Python. The distribution of the continuous variables was checked for normality before the analysis. Then, partial correlation coefficients were estimated to examine the relationships between academic achievement, CRF and executive functions, controlling for age, by gender.

Because there are not universally accepted cut offs for categorizing low, middle and high CRF and executive performance, we used quartiles for this aim as follows: low (Q1), middle (Q1-Q2) and high (Q4). Using these categories, ANCOVA models were estimated to examine differences in academic achievement (language and mathematics) by cardiorespiratory fitness and executive function performance (inhibition, cognitive flexibility and working memory). These ANCOVA models were also used to examine the potential confounding or mediating role of executive functions in the relationship between CRF and academic achievement. In these models, language and mathematics grades were used as dependent variables and weight status, CRF, inhibition, cognitive flexibility and working memory categories as fixed factors. In these models, language and mathematics were used as dependent variables and CRF, inhibition, cognitive flexibility and working memory categories as fixed factors, controlling for age and mother educational level (Model 1). Further adjustment for the executive function index or CRF, depending on the fixed factor, was included in Model 2. Pairwise post hoc hypotheses were tested using the Bonferroni correction for multiple comparisons. Effect size partial eta$^2$ for ANCOVA test was calculated. Indicates small effects values of 0.01, intermediate effects 0.06 and strong effects > 0.14 [26]. These ANCOVA models were estimated for the entire sample and stratified by gender. For the rest of the analyses, continuous variables were used.

Mediation analysis was conducted to assess whether executive function dimensions were mediators in the relationship between CRF and academic achievement. For these analyses, we used PROCESS SPSS Macro version 3.1, and 5.000 bias-corrected bootstrap samples [27]. For these analyses, we used PROCESS SPSS Macro version 3.1, selecting Model 4 and 5.000 bias-corrected bootstrap samples [27]. Mediation was assessed by the indirect effect of CRF (independent variable) on academic achievement variables (language and mathematics, individually entered as dependent variables) through executive functions (inhibition, cognitive flexibility and working memory, individually entered as mediators). The total (c path), direct (c′ path), and indirect effect (a*b paths) coefficients, as well as indirect effects (IE) with 95% confidence intervals, were presented, regardless of the significance of the total effect (CRF on academic achievement) and direct effect (the effect on academic achievement when CRF and executive functions are included as predictors) [27]. Simple mediation models were tested, by introducing independently the three different mediating variables. Then, a parallel serial mediation model was tested in which the three domains of executive function (inhibition, cognitive flexibility and working memory) sere simultaneously entered in the model in such a way that each mediator was controlled for the others. The percentage of mediation ($P_M$) was calculated as (indirect effect/total effect) x 100 to estimate the percentage of the total effect explained by the mediation path, whenever the total effect was larger than the indirect effect and in the same direction [27].

All analyses were adjusted by age and mother educational level.

## Results

Our study included 570 schoolchildren (mean age = 9.58 years, SD = 0.72), of whom 279 (51.7%) were girls. The descriptive characteristics of the sample are included in Table 1. There were several differences between gender in CRF, reaction time in inhibition, cognitive flexibility and language, were higher for girls (p < 0.05).

The partial correlation coefficients between academic achievement subjects, CRF and executive function domains are presented in Table 2. CRF was positively correlated with language, mathematics, inhibition and cognitive flexibility in both genders and was positively correlated with working memory in boys (p < 0.05), but the correlation coefficient between CRF and working memory was not significant in girls. In addition, positive correlations were found between academic achievement and all executive functions in both boys and girls (all p < 0.001). See also S1 Fig.

Mean differences in language and mathematics (by CRF) and executive functions (by gender), are shown in Figs 1 and 2 and S1 and S2 Tables in the supplementary material. The mean grades in both language and mathematics were significantly higher in children in the higher categories of CRF and executive function domains (p < 0.05), except for mean grades by CRF categories in boys and language by inhibition categories in girls, in which statistical significance was not achieved. Overall, differences in academic achievement maintained their statistical significance in Model 2, except for mean grades in mathematics and inhibition categories in girls (p = 0.194).

The mediation model scheme is displayed in Fig 3, and the results of the mediation analysis are shown in Table 3. Inhibition and cognitive flexibility mediated the association between CRF and academic achievement, controlling for age and mother educational level. In the first regression step (path a), CRF was positively related to the executive function domains (p < 0.05). In the second step (path c), the regression coefficients of CRF were positively associated with academic achievement in all mediation models (p < 0.001). In the last regression model (path b), the executive function domains were positively related to the dependent variable (p < 0.001), and the relationship between CRF and academic achievement became weaker (although significant) when the executive function domains were included in the model (path c'). Finally, the indirect effects were statistically significant in for inhibition and cognitive flexibility, confirming that executive function domains acted as a partial mediator in these models, except for working memory. The mediation analyses by gender showed that, after controlling for mother educational level, cognitive flexibility was not significant in girls and inhibition in boys. When parallel serial mediation was performed, the results were similar (S2 Fig).

## Discussion

The results of this study reveal that children in higher CRF and executive function categories had better academic achievement than their peers in other categories. Furthermore, our study suggested that the association between CRF and academic achievement in mathematics and language was partially mediated by inhibition and cognitive flexibility, even after controlling for age and mother educational level. However, the association between CRF and academic achievement was not mediated by working memory.

Several studies support that executive functions underlie the selection, planning and monitoring of goal-directed processes [1], which are involved in more complex processes such as perception, attention or memory [8]. Thus, executive function provides neural substrates that underlie successful learning [28]. Studies have also reported that children with higher levels of physical fitness perform better cognitively [13], suggesting that motor and cognitive development may be interrelated [29]. Increasingly conclusive data support that physical fitness

**Table 1. Characteristics of the study sample, by gender.**

| | Total (n = 570) | Boys (n = 270) | Girls (n = 293) | p |
|---|---|---|---|---|
| Age (years, months) | 9.58 (0.72) | 9.55 (0.74) | 9.62 (0.70) | 0.252 |
| Height (cm) | 140.93 (7.33) | 141.03 (7.09) | 140.84 (7.56) | 0.766 |
| Weight (Kg) | 36.64 (10.02) | 36.82 (10.22) | 36.48 (9.84) | 0.692 |
| BMI (Kg/m$^2$) | 18.26 (3.88) | 18.33 (0.25) | 18.19 (0.22) | 0.672 |
| Tanner Stage | 1.64 (0.80) | 1.58 (0.69) | 1.68 (0.86) | 0.288 |
| CRF (20 m shuttle run; VO$_{2max}$) [*] | 45.96 (4.61) | 47.64 (4.99) | 44.66 (3.81) | **<0.001** |
| Low | 40.71 (1.54) | 41.08 (1.75) | 40.45 (1.31) | |
| Middle | 45.93 (2.37) | 47.18 (2.46) | 44.58 (1.30) | **<0.001** |
| High | 51.82 (2.81) | 54.13 (1.76) | 49.87 (1.91) | |
| CRF (20 m shuttle run; stages) [f] | 3.77 (1.86) | 4.32 (2.04) | 3.27 (1.51) | **<0.001** |
| Family SES (n; %): | 103, 18.3% | 51, 18.9% | 52, 17.7% | 0.696 |
| Lower / Lower middle | 250, 44.4% | 121, 44.8% | 129, 44% | |
| Middle | 130, 23.1% | 56, 20.8% | 74, 25.2% | |
| Upper middle / Upper | | | | |
| Mother educational level (n; %): | | | | |
| No reading/No studies | 9 (1.8%) | 1 (0.4%) | 8 (1.1%) | |
| Elementary | 53 (10.6%) | 27 (11.3%) | 26 (9.9%) | |
| Secondary studies | 221 (44.1%) | 102 (42.9%) | 119 (45.2%) | 0.265 |
| High school | 134 (26.7%) | 66 (27.7%) | 68 (25.9%) | |
| University studies | 84 (16.8%) | 42 (17.6%) | 42 (16%) | |
| Academic achievement: | | | | |
| Language | 7.18 (1.72) | 7.03 (1.76) | 7.32 (1.67) | **0.046** |
| Mathematics | 6.90 (1.87) | 6.98 (1.82) | 6.82 (1.91) | 0.303 |
| Executive function: | | | | |
| Inhibition (FT) | | | | |
| ■ Total[¥] | 7.62 (1.07) | 7.58 (1.16) | 7.67 (0.98) | 0.344 |
| ■ Accuracy[$] | 19.69 (1.58) | 19.62 (1.36) | 19.70 (1.33) | 0.491 |
| ■ TR[±] congruent | 0.84 (0.26) | 0.85 (0.29) | 0.82 (0.22) | 0.245 |
| ■ TR[±] incongruent | 1.02 (0.26) | 1.05 (0.47) | 1.00 (0.37) | 0.145 |
| Cognitive flexibility (DCST) | | | | |
| ■ Total[§] | 7.01 (1.11) | 6.95 (1.24) | 7.16 (1.08) | 0.270 |
| ■ Accuracy[$] | 28.17 (3.10) | 27.88 (3.10) | 28.44 (3.08) | **0.036** |
| ■ TR[±] switch | 0.99 (0.29) | 1.02 (0.30) | 0.95 (0.27) | **0.003** |
| ■ TR[±] non-switch | 0.91 (0.27) | 0.94 (0.29) | 0.88 (0.24) | **0.009** |
| Working memory (LSWM)[Δ] | 14.26 (3.18) | 14.47 (3.32) | 14.06 (3.04) | 0.136 |

Values are mean ± standard deviation (SD), except for family socioeconomic status (SES), which is shown as percentages (n: %).

Abbreviations: BMI = body mass index; CRF = cardiorespiratory fitness; FT = Flanker Task; RT = Reaction Time; DCST = Dimensional Change Card Sort Test; LSWM = List Shorting Working Memory.

Bold text indicates statistical significance of p < 0.05.

[*] Last one-half stage measured in 20-m shuttle run test (1 stage = 1 minute).

[f] According to Léger recommendations.

[¥] Total score was calculated using a two-vector method that incorporates both accuracy and time reaction [23]. Total score = (0.25 x number correct responses) + 5 – LOG$_{10}$ [(time reaction congruent + time reaction incongruent / 2)]. Range 0–10.

[§] Total score was calculated using a two-vector method that incorporates both accuracy and time reaction [23]. Total score = (0.167 x number correct responses) + 5 – LOG$_{10}$ [(time reaction congruent + time reaction incongruent / 2)]. Range 0–10.

[Δ] Total score: correct items across trials.

[$] Accuracy is number of correct answers in each test.

[±] Reaction Time in milliseconds.

**Table 2. Partial correlation coefficients among CRF, executive functions and academic achievement subjects controlling for age and mother educational level, by gender.**

|  |  | BMI | CRF | Language | Mathematics | Inhibition | Cognitive flexibility |
|---|---|---|---|---|---|---|---|
| CRF | Total | -.472** | - |  |  |  |  |
|  | Boys | -.538** |  |  |  |  |  |
|  | Girls | -.467** |  |  |  |  |  |
| Language | Total | -.043 | .204** | - |  |  |  |
|  | Boys | -.055 | .158* |  |  |  |  |
|  | Girls | -.032 | .338** |  |  |  |  |
| Mathematics | Total | -.051 | .215** | .830** | - |  |  |
|  | Boys | -.080 | .146* | .857** |  |  |  |
|  | Girls | -.033 | .276** | .826** |  |  |  |
| Inhibition | Total | -.065 | .141* | .330** | .335** | - |  |
|  | Boys | -.029 | .102 | .386** | .439** |  |  |
|  | Girls | -.102 | .227* | .268** | .248** |  |  |
| Cognitive flexibility | Total | -.111* | .129* | .341** | .332** | .493** | - |
|  | Boys | -.086 | .163* | .324** | .333** | .566** |  |
|  | Girls | -.133* | .138* | .361** | .353** | .442** |  |
| Working memory | Total | .023 | .059 | .283** | .297** | .315** | .283** |
|  | Boys | -.040 | .056 | .220* | .283** | .318** | .271** |
|  | Girls | .067 | .011 | .358** | .298** | .325** | .305** |

Abbreviations: CRF = cardiorespiratory fitness, $VO_{2max}$.

P value =

* $p < 0.05$;

** $p < 0.001$

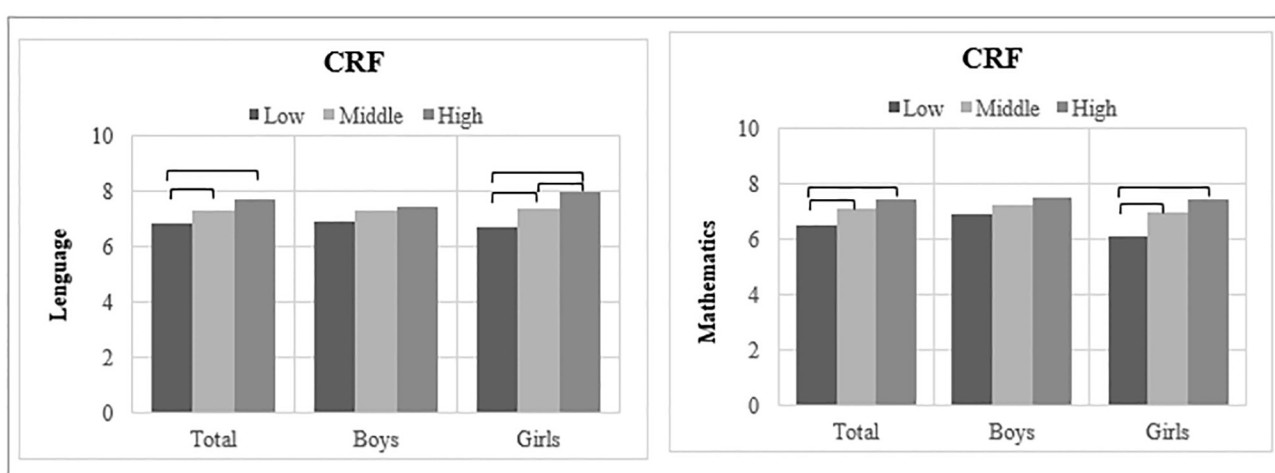

**Fig 1. Mean differences in academic achievement (language and mathematics) by CRF, inhibition, cognitive flexibility and working memory categories.** Brackets indicate significant differences in mean statistical significance ($p < 0.05$) between categories in the Bonferroni multiple comparison post-hoc test. Abbreviations: CRF = cardiorespiratory fitness. CRF categories indicate quartiles: low (Q1), middle (Q2-Q3) and high (Q4).

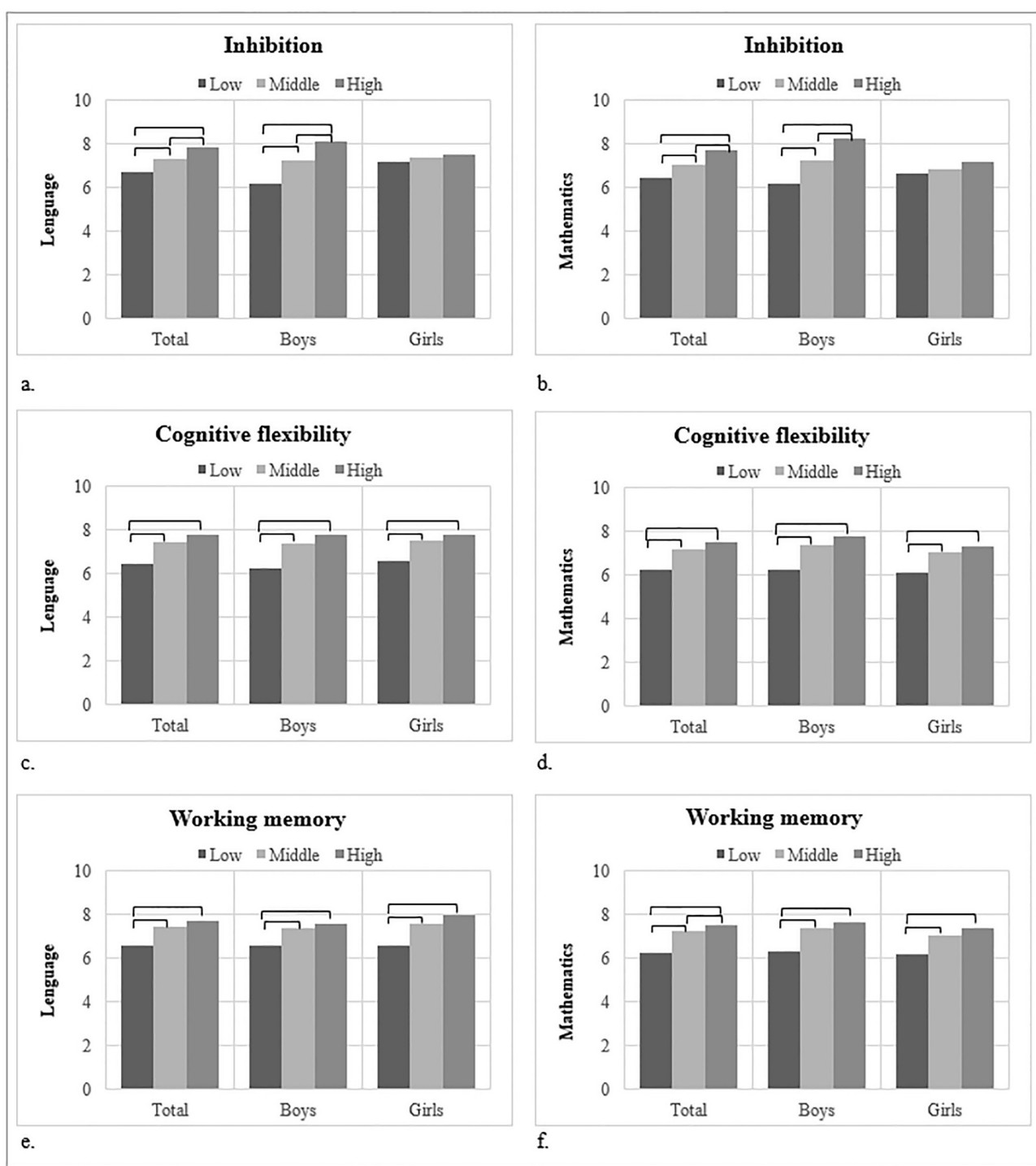

**Fig 2. Mean differences in academic achievement (language and mathematics) by CRF, inhibition, cognitive flexibility and working memory categories.** Brackets indicate significant differences in mean statistical significance (p < 0.05) between categories in the Bonferroni multiple comparison post-hoc test. Abbreviations: CRF = cardiorespiratory fitness. CRF categories indicate quartiles: low (Q1), middle (Q2-Q3) and high (Q4).

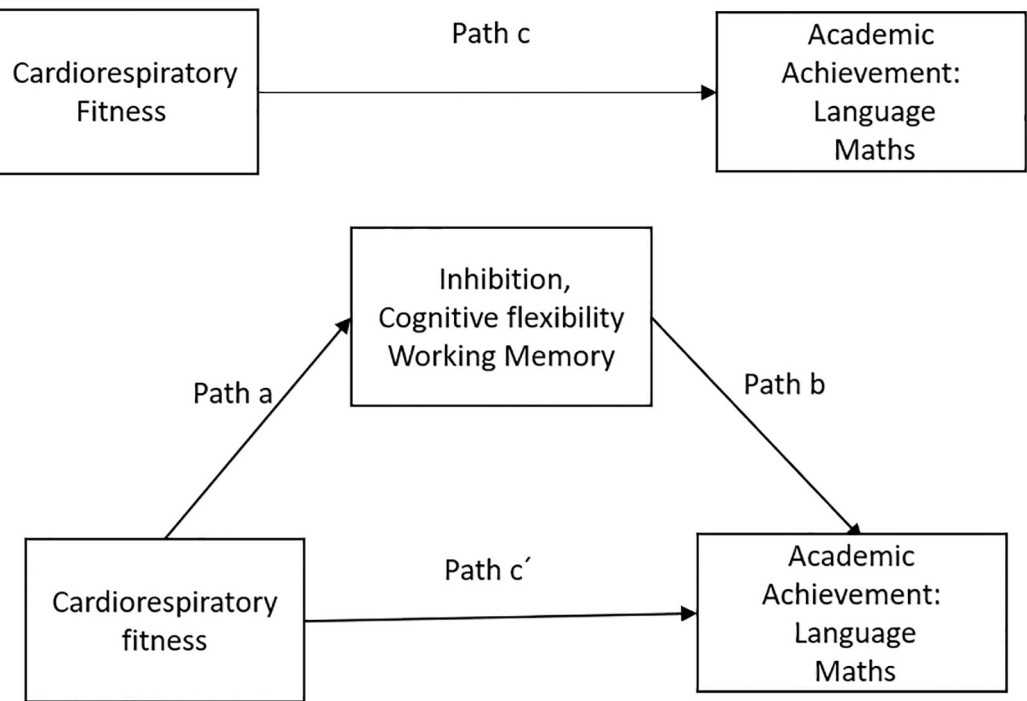

**Fig 3. Mediation model.** Cardiorespiratory fitness (independent variable) on academic achievement (dependent variable), through executive functions (mediator variable). Path a, association between independent variable and mediators; path b, association between mediator and dependent variables; path c, total effect between independent and dependent variables; path c´, direct effect of independent variable on dependent variable.

induces vascularization, neural growth and synaptic transmission in the regions of the brain underlying executive functions, especially the prefrontal cortex [1,30].

Our data, similar to those of previous studies [5,14,15], support this line of argumentation since they show a positive association between executive functions and academic achievement and between CRF and academic achievement. Moreover, our data suggest that the influence of CRF on academic achievement is mediated through some cognitive domains of executive function, such as inhibition and cognitive flexibility. These results are in line with those of other authors [14–16] who tested indirect paths, suggesting that physical fitness is not an independent predictor of academic achievement but acts via executive functions.

The results of studies analyzing the potential mediator role of executive function in the relationship between fitness and academic achievement are not conclusive. Van der Niet et al. reported a significant indirect relationship through executive functions (measured as a latent variable composed of problem solving skills and cognitive flexibility) between a physical fitness index and overall academic achievement [14]. Similarly, Kvalo et al. reported a mediating role of executive functions in mathematics but not in reading or English [16]. The results of the study by De Bruijn et al., showed that the mediating role of executive function is not general but specific to the academic achievement domain such that verbal working memory acts as a mediator between physical fitness (measured using the EUROFIT battery) [31] and mathematics and spelling, whereas visuospatial working memory acts as a mediator only in mathematics [15]. The differences in the measurement methods of the three constructs could be the cause of these inconsistencies since, for example, most studies tested models in which physical fitness was measured as an index that included both CRF and strength, despite the evidence that supports different pathways for the influence of these fitness components on cognition.

**Table 3. Total, direct, and indirect effects, a and b pathways, of the simple mediation analyses investigating executive functions as a mediator between cardiorespiratory fitness and academic performance, controlled by age and mother educational level.**

| Y | Mediator | Total Effect (c) | Direct Effect (c′) | Path a | Path b | Indirect effect (ab) | 95% CI | | $P_M$ (%) |
|---|---|---|---|---|---|---|---|---|---|
| | | | | | | | Lower | Upper | |
| Total | | | | | | | | | |
| Language | Inhibition | .074** | .058** | .035 ** | .450 ** | .015 | .005 | .028 | 20.27 |
| | Cognitive flexibility | .074 ** | .059 ** | .032 * | .471 ** | .015 | .003 | .028 | 20.27 |
| | Working memory | .073 ** | .068 ** | .042 | .140 ** | .006 | -.005 | .018 | |
| Mathematics | Inhibition | .081 ** | .064 ** | .035 ** | .472 ** | .016 | .005 | .030 | 19.75 |
| | Cognitive flexibility | .081 ** | .066 ** | .032 * | .473 ** | .015 | .003 | .029 | 18.51 |
| | Working memory | .080 ** | .074 ** | .042 | .152 ** | .006 | -.005 | .020 | |
| Boys | | | | | | | | | |
| Language | Inhibition | .054 * | .041 | .025 | .520 ** | .013 | -.003 | .032 | 24.07 |
| | Cognitive flexibility | .054 * | .037 | .033 * | .523 ** | .017 | .003 | .037 | 31.48 |
| | Working memory | .054 * | .050 * | .036 | .114 ** | .004 | -.004 | .009 | |
| Mathematics | Inhibition | .050 * | .035 | .025 | .583 ** | .015 | -.004 | .036 | 30.00 |
| | Cognitive flexibility | .050 * | .032 | .033 * | .531 ** | .018 | .003 | .038 | 36.00 |
| | Working memory | .050 * | .044 | .036 | .145 ** | .005 | -.010 | .021 | |
| Girls | | | | | | | | | |
| Language | Inhibition | .143 ** | .124 ** | .062 ** | .313 * | .019 | .004 | .041 | 13.38 |
| | Cognitive flexibility | .143 ** | .125 ** | .044 * | .425 ** | .019 | -.003 | .040 | |
| | Working memory | .143 ** | .141 ** | .009 | .176 ** | .002 | -.021 | .029 | |
| Mathematics | Inhibition | .128 ** | .108 ** | .062 ** | .327 ** | .020 | .005 | .044 | 15.62 |
| | Cognitive flexibility | .128 ** | .108 ** | .044 * | .464 ** | .021 | -.004 | .045 | |
| | Working memory | .127 ** | .126 ** | .009 | .160 ** | .002 | -.018 | .027 | |

Results showed unstandardized coefficients (standard error) and BC 95% CI based on 5000 bootstraps. All analyses were adjusted for age and mother educational level.

CRF cardiorespiratory fitness; CI confidence interval.

* $p < 0.05$;

** $p < 0.005$.

Our study found a mediator role of cognitive flexibility and inhibition, but contrary to previous studies [15], the potential mediation effect of working memory in the relationship between CRF and academic achievement was not confirmed. Developmental variability in children's executive function could be the cause of this contradictory finding: although executive functions develop from infancy to adulthood, the age at which each executive function is stabilized is still matter of debate [32]. Some studies have suggested that while inhibition stabilizes by the school years, working memory continues to develop until late adolescence, and cognitive flexibility is the latest maturing cognitive function [32]. Moreover, despite the apparent similarities of the different tasks used to measure each executive function, different conflict tasks show different ages of mastery, indicating different cognitive demands [32].

Our results indicated that the mediator effect of EF between CRF and academic achievement was approximately 15 to 36%. Several other fitness-related factors could influence academic achievement, such as motor skills. Two studies showed that motor skills, rather than aerobic fitness or physical activity, had the strongest association with executive functions and academic performance [33,34]. A review study [35] proposed that both CRF and motor skills were related to academic performance and that the influence of both could affect cognition and academic performance in different ways. Moreover, a mediation study showed that motor coordination—not cardiovascular endurance or muscular strength—was the only motor

ability that accounted for a significant percentage of the variance in academic achievement [36]. Other variables that have been demonstrated to be related to academic achievement, possibly influencing the relationship between CRF and academic achievement, are physical activity and general health improved by cardiorespiratory fitness [18].

In this study, we reported slight differences between boys and girls in the relationship between CRF and academic achievement and in the mediator role of executive functions. We found higher values for girls in language and cognitive flexibility and a stronger association of fitness with academic achievement in girls compared to boys. Previous studies have reported differences in brain maturation and brain structures [18], which may explain the higher cognitive flexibility in girls because they reach a plateau before boys [18] in many executive functions, especially cognitive flexibility, which is the executive function that matures latest [32]. However, mediation analyses revealed a stronger mediating role of executive functions in the relationship between fitness and academic achievement in mathematics in boys than in girls after controlling by mother educational level. This might be due to the higher influence of mother educational level on girls than on boys.

## Strengths and limitations

The strengths of this study include the large sample size and the fact that it analyzed the mediating effect of different domains of executive functions in the association between CRF and academic achievement. Mediation analysis has been proposed as a statistical procedure suitable for clarifying whether a third variable underlies the relationship between two variables. Thus, this procedure allows us to consider the significant proportion of the influence of the independent variable (CRF) on the dependent variable (academic achievement) takes place through executive functions (mediator) [37].

However, this study had some limitations. Cross-sectional studies utilizing statistical mediation provide a theoretical and mechanistic foundation about the relationships between cardiorespiratory fitness, brain, and cognition. Nevertheless, their correlational nature leaves open the possibility that the observed behavioral and structural fitness-related differences between high and low-fit groups are caused by some unmeasured factor. Furthermore, although previous studies support a directional relationship between physical activity and fitness to academic achievement, the cross-sectional nature of our analyses prevented us from making cause–effect inferences, since temporal ambiguity bias cannot be completely ruled out. For that all, randomized controlled trials are necessary to account for potential selection bias, as well as to establish a direct, causal relationship in humans between aerobic fitness, brain structure, and cognitive functioning. Second, the evaluation of a single fitness component in our study makes it difficult to perform comparisons with other studies, where fitness has been assessed with another test or with a global physical fitness index. Another limitation of the study was that both CRF and academic performance were not assessed simultaneously, and they were assessed using indirect procedures, i.e., the 20-meter shuttle run test and teacher evaluations. We should consider that academic achievement is the result of a long-term learning process, and the cross-sectional evaluation actually implies the final outcome of a 1-year learning follow up. Regarding the validity of the 20-meter shuttle run test to estimate CRF, this is the recommended test in the ALPHA battery, since the direct measurement of $VO_{2max}$ in population-based studies is unfeasible. Finally, another point to mention is the inherent difficulties in the measurement of executive functions, so-called task impurity [15,32]; many of the tasks used to measure an executive function also require involvement from the rest of the executive functions. For this reason, some authors recommend including several tasks for each executive function to obtain a more reliable measure of executive

functioning [29]. Moreover, because some authors recommend the measurement of accuracy and reaction time as separate variables, the assessment of these cognitive abilities as a unique score could be seen as a limitation; however, the NIH Toolbox´s total score has been used and validated in several studies and allows the avoidance of a ceiling effect in accuracy, which is typically achieved by most participants of this age.

## Conclusions

This study supports the positive associations between academic achievement and CRF and between academic achievement and executive functioning in schoolchildren. Additionally, this study shows that mental processes such as inhibition and cognitive flexibility mediate the relationship between CRF and academic achievement in language and mathematics. However, working memory does not mediate this association. The observations from these data highlight the importance of promoting CRF in children to improve academic achievement. The correlations between executive functions and academic achievement suggest that it may be beneficial to target them to facilitate adequate learning in schools. Thus, the design of physical activity programs aimed at increasing CRF and executive functions might be a practical strategy to improve academic achievement in schoolchildren.

## Supporting information

**S1 Fig.**
(TIFF)

**S2 Fig.**
(TIF)

**S1 Table. Mean differences (ANCOVA) in academic achievement in language and mathematics by CRF categories, controlling for age and mother educational level.**
(DOCX)

**S2 Table. Mean differences (ANCOVA) in academic achievement in language and mathematics by inhibition, cognitive flexibility and working memory categories, controlling for age and mother educational level.**
(DOCX)

## Acknowledgments

We thank the schoolchildren, teachers, schools and families for their collaboration and participation in the study.

## Author Contributions

**Conceptualization:** Mairena Sánchez-López, Vicente Martínez-Vizcaíno, Marta Nieto-López.

**Data curation:** María Eugenia Visier-Alfonso, Estela Jiménez-López, Andrés Redondo-Tébar.

**Formal analysis:** Vicente Martínez-Vizcaíno.

**Funding acquisition:** Mairena Sánchez-López, Vicente Martínez-Vizcaíno.

**Investigation:** María Eugenia Visier-Alfonso, Mairena Sánchez-López, Vicente Martínez-Vizcaíno, Estela Jiménez-López, Andrés Redondo-Tébar.

**Methodology:** María Eugenia Visier-Alfonso, Mairena Sánchez-López.

**Project administration:** Vicente Martínez-Vizcaíno.

**Software:** María Eugenia Visier-Alfonso.

**Supervision:** Marta Nieto-López.

**Writing – original draft:** María Eugenia Visier-Alfonso.

**Writing – review & editing:** Mairena Sánchez-López, Vicente Martínez-Vizcaíno, Andrés Redondo-Tébar, Marta Nieto-López.

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
