## [Decision Letter · Decision Letter 0]

18 Jul 2019

PONE-D-19-14578

Att

Executive functions as a mediator in the relationship between cardiorespiratory fitness and academic achievement in schoolchildren.

PLOS ONE

Dear Mrs. SANCHEZ,

Thank you for submitting your manuscript to PLOS ONE. After careful consideration, we feel that it has merit but does not fully meet PLOS ONE’s publication criteria as it currently stands. Therefore, we invite you to submit a revised version of the manuscript that addresses the points raised during the review process.

The manuscript has been assessed by three reviewers; their comments are available below.

The reviewers have raised some concerns which need attention in a revision. The reviewers note that further clarification and justification should be provided for methodological aspects of the study, including additional information on the sampling approach, inclusion and exclusion criteria and details on how each measurement was carried out. The reviewers also note that given the cross-sectional design, you should carefully review the language and ensure that any statement that make causal relationships are revised to only refer to associations.

Could you please carefully revise the manuscript to address the concerns raised by the reviewers?

We would appreciate receiving your revised manuscript by Aug 31 2019 11:59PM. Please include the following items when submitting your revised manuscript:

We look forward to receiving your revised manuscript.

Kind regards,

Iratxe Puebla

Senior Managing Editor, PLOS ONE

Journal Requirements:

1. There is some discrepancy between your manuscript and the details registered at www.clinicaltrials.gov, please clarify whether schoolchildren included were aged 8 to 11 years or 9 to 11 years, and whether the number of children enrolled in the study was 570 or 563.

2. Please amend either the title on the online submission form (via Edit Submission) or the title in the manuscript so that they are identical.

Reviewers' comments:

Reviewer's Responses to Questions

**Comments to the Author**

1. Is the manuscript technically sound, and do the data support the conclusions?

Reviewer #1: Yes

Reviewer #2: Partly

Reviewer #3: Yes

2. Has the statistical analysis been performed appropriately and rigorously? 

Reviewer #1: Yes

Reviewer #2: No

Reviewer #3: Yes

3. Have the authors made all data underlying the findings in their manuscript fully available?

Reviewer #1: Yes

Reviewer #2: Yes

Reviewer #3: Yes

4. Is the manuscript presented in an intelligible fashion and written in standard English?

Reviewer #1: Yes

Reviewer #2: Yes

Reviewer #3: Yes

5. Review Comments to the Author

Reviewer #1: The investigators performed a cross-sectional analysis of the baseline data from a cluster-randomized controlled trial , aimed at assessing if the effectiveness of a physical activity program diminishes cardiometabolic risk and improves CRF, executive functioning and academic achievement in schoolchildren. Assuming the sample to have been adequately selected, the overall goal of examining physical fitness influences on academic achievement through executive functions appears to have been achieved. Although not a randomized comparison of low vs. high CRF, the descriptive nature of the study using correlational analysis, the ANCOVA with post hoc adjustment and the mediation analysis, is certainly interesting. The mediation model is well utilized and the results are well explained. There are some minor concerns.

1. The manuscript should be edited for typos. For example on line 26 of the abstract, the expression ,’ Cross-sectional study including 539 schoolchildren, aged 8 to 11 years, from Cuenca, Spain.’ Is not a full sentence. On Table 3, one assumes the word ‘Hight’ should be ‘High’.

2. On Table 1 the significant (p<0.05) p-values according to line 181 are in bold. Why are the p-values for the executive functions, TR congruent, incongruent and Total (0.245, 0.145 and 0.270, respectively) in bold type?

Reviewer #2: Dear editor,

The paper focus on an important and emergent research area. However, in my opinion, deep changes on the manuscript should be considered prior to acceptance.

Abstract:

The authors should present some statistical data related to the main results, not only a p value.

The abstract readability can be improved by a scientific language edition (e.g: line 30 -31 change from “…showed significantly higher mean scores…” to “… higher scores…”).

Line 34: What means partially mediated?

The last sentence from the abstract is merely speculative and it is not related to the study conclusions.

How was the sampling process conducted? Please provide the inclusion and exclusion criteria.

Do the authors have formal assent from the children?

What is the time gap between academic performance data and CRF and cognitive functions assessments?

Line 107 Please change VO2max to VO2max

Line 109: What app was used to assess executive functions? Please change from ipad to tablet. Describe model and manufacturer in parentheses.

It is mandatory to the authors to provide more information about the measurements of academic performance.

Mother educational level is one of the main predictor of children academic performance. The authors should used this data as a controlling variable instead of a SES index, otherwise, the authors must present a better rationally for using this index.

Why partial correlations were not adjusted for SES?

Why the authors opt for using quartiles?

Did the authors calculated a priori sample size? What is the statistical power of the analyses?

I am not an expert in statistical, but I am not confident the authors made the right choices.

The authors should consider to present the data with the level of accuracy from the instrument used to measure the outcomes (e.g: there is no decimal places for age in years).

I suggest the authors to avoid the VO2 calculation using equation; they can simple use the last complete stage on shuttle run test.

What is the criteria for SES categorization?

Would be of value if the authors should use some data related to body composition in the analyses (at least a BMI could be presented).

The results section is hard to flow. Firstly, I believe the authors should rethink their statistical procedures. I strongly suggest the authors to avoid categorization and use continuous data.

The correlations may be presented in figures.

If the authors accept my suggestion and change the statistical approach, the discussion should be rewritten in the light of the new results.

For me the main limitations are: indirect assessment of CRF and academic performance; If academic performance and CRF are not assess in the same time point (as I am assuming), it could also be a strong limitation of the present investigation.

Reviewer #3: Review PONE-D-19-13578

Excecutive functions as a mediator in the relationship between cardiorespiratory fitness and academic achievement in schoolchildren.

Comments

Title: can the title be changed to a declarative statement (e.g. Executive functions mediate the relationship between…) and better specify the study population (add country)?

Abstract:

“It has been hypothesized that” or previous researched have actually studied this relationship?

“the associations between cardiorespiratory fitness (CRF) and the executive function domains with academic achievement and…” is unclear. Associations between A and B with C? Do you mean interrelationships (e.g. A and B, A and C, B and C)?

Line 26 – Add “This” before cross-sectional.

Can you add a little more detail on the measurement of CRF and EF? Just add shuttle-run and NIH Toolbox.

Can you add effect sizes to the abstract?

Line 36 – this sentence errs on the causal side and; would reword to state that children in this study with higher CRF and EF had higher math and language scores. It’s just an association in this sample.

Line 38 – mediated (past tense); otherwise it sounds like a population inference

Introduction

You may also want to cite

• Scott et al. 2016, Med Sci Sport Exer – Cardiorespiratory fitness is associated with better executive function in young women

• Scott et al. 2016, Journal of Nutrition – Combined iron deficiency and low aerobic fitness doubly burden academic performance among women attending university (see fig 3 for mediation)

Line 73 – same problem as in abstract: Associations between A and B with C?

Methods

Line 98 – “in standardized conditions”? Do you mean “following standard procedures”? Was this done by nurses or the researchers?

20 trials on the flanker task does not seem sufficient. Were multiple blocks performed? If you used only 20 trials, can you cite other studies to support that this is acceptable?

Would it not be useful to treat accuracy and speed as separate variables? Were children with <80% accuracy dropped from the analysis?

Line 132 – what are aliments? Food items?

Please provide more information on the SES index. What were the education and employment categories and how were these used to construct the SES index? You say highest level of education in their family (all family members) and then either father or mother. Which was it? Too bad you didn’t ask about their income.

The model descriptions are unclear to me. All of your variables are continuous, so why not just use OLS regressions with multiple predictors? “We estimated ANCOVA models to test mean differences in language and mathematics by categories of CRF and executive function domain, controlling for age, pubertal status and SES (Model 1); further adjustment for the executive function index or CRF, depending on the fixed factor, was included in Model 2.”

Does adjusting for pubertal status (I think you mean stage; status implies a dummy variable, i.e. pre/post pubertal) really add much? You already adjust for age, which I would assume is colinear with pubertal status, though I could be wrong.

Results

Table 1

-add BMI

-add % falling into fitness categories based on CRF? VO2max of 46 seems fairly fit, but I don’t know what the ACSM cutoffs are in this age group. Need to get a sense of whether sample is fit or unfit.

-need units for EF indicators. I assume accuracy is % incorrect responses? This is typically expressed as proportion correct.

Table 3 is very hard to follow. I suggest moving this to supplement and creating a figure or figures for the main paper.

Discussion – can you comment on what may mediate the other 70-85% of the relationship between CRF and language/math that is not explained by executive function, age, pubertal satus, and SES?

6. PLOS authors have the option to publish the peer review history of their article (what does this mean?). If published, this will include your full peer review and any attached files.

Reviewer #1: No

Reviewer #2: Yes: Wagner Luiz do Prado

Reviewer #3: No

---

## [Author Response · Author response to Decision Letter 0]

3 Sep 2019

Response to Reviewer 1 Comments

The investigators performed a cross-sectional analysis of the baseline data from a cluster-randomized controlled trial, aimed at assessing if the effectiveness of a physical activity program diminishes cardiometabolic risk and improves CRF, executive functioning and academic achievement in schoolchildren. Assuming the sample to have been adequately selected, the overall goal of examining physical fitness influences on academic achievement through executive functions appears to have been achieved. Although not a randomized comparison of low vs. high CRF, the descriptive nature of the study using correlational analysis, the ANCOVA with post hoc adjustment and the mediation analysis, is certainly interesting. The mediation model is well utilized, and the results are well explained. There are some minor concerns.

Authors’ response: Your comments and general involvement in the review process are highly appreciated. We have addressed all concerns, and we hope our revisions have improved the manuscript.

1. The manuscript should be edited for typos. For example, on line 26 of the abstract, the expression,’ Cross-sectional study including 539 schoolchildren, aged 8 to 11 years, from Cuenca, Spain.’ Is not a full sentence. On Table 3, one assumes the word ‘Hight’ should be ‘High’.

Authors’ response: Thank you for the comment. We apologize for the error. The paper has been sent again to a specialist service to correct mistakes and typos. The following changes have been made:

 “This was a cross-sectional study that included 570 schoolchildren, aged 8 to 11 years, from Cuenca, Spain” (Page 2, lines 32).

Tables S1 and S2 (S1 and S2 Table Supplementary material).

2. On Table 1 the significant (p<0.05) p-values according to line 181 are in bold. Why are the p-values for the executive functions, TR congruent, incongruent and Total (0.245, 0.145 and 0.270, respectively) in bold type?

Authors’ response: Thank you for the comment. We apologize for the mistake, which has been corrected (Table 1, page 10).

Response to Reviewer 2 Comments

The paper focus on an important and emergent research area. However, in my opinion, deep changes on the manuscript should be considered prior to acceptance.

Authors’ response: Your comments and general involvement in the review process are highly appreciated. We have addressed all concerns, and our revisions have improved the manuscript.

Abstract:

The authors should present some statistical data related to the main results, not only a p value.

Authors’ response: We appreciate this comment. As suggested, we have included the following information in the abstract section:

“Overall, ANCOVA models controlling for age, gender and mother educational level showed higher scores in language and mathematics in children in higher categories of CRF, inhibition, cognitive flexibility and working memory than in children in lower categories. The effect sizes were moderate (p < 0.05, partial eta squared: from 0.05 to 0.12). Moreover, the mediation analysis showed that inhibition partially mediated the relationship between CRF and language (c’ = 0.058; IC = [0.005; 0.028]) and mathematics (c’ = 0.064; IC = [0.005; 0.030]) grades; similarly, cognitive flexibility mediated CRF’s relationship with language (c’ = 0.059; IC = [0.003; 0.028]) and with mathematics (c’ =0.066. ; IC = [0.003; 0.029]); however, a significant relationship remained. For working memory, mediation analysis showed no significant results (c′ = 0.92; IC= [-0.002;0.025] P > 0 .05 in language; c′ = 0.103; IC= [-0.002;0.029] P > 0.05 in mathematics). Mediation ranged from 13.38 % to 36 %” (Page 2, lines 43-50). 

The abstract readability can be improved by a scientific language edition (e.g: line 30 -31 change from “…showed significantly higher mean scores…” to “… higher scores…”).

Authors’ response: Thank you for the comment. As suggested, we have rephrased that sentence in the abstract section to improve readability:

“ANCOVA models controlling for age, gender and mother educational level showed higher scores in language and mathematics in children (…)” (Page 3, lines 37-38).

Moreover, full abstract has been improved following the comments and suggestions.

Line 34: What means partially mediated?

Authors’ response: According to Andrew F. Hayes (Hayes AF. Introduction to mediation, moderation, and conditional process analysis: a regression-based approach.: Guildford Press; 2018), partial mediation implies that the mechanism through the mediator variables (executive functions) does not entirely account for the association observed between the independent variable (CRF) and the dependent variable (academic achievement in language and mathematics), whereas complete mediation means that the association between CRF and academic achievement is entirely accounted for by the indirect mechanism. Partial mediation is the case in which the path from the independent variable to the dependent variable is reduced in absolute size but is still different from zero when the mediator variable is introduced in the model, whereas complete mediation occurs when the relationship between the independent variable and the dependent disappears after entering the mediator in the model, making the path c´ = 0. 

In order to clarify this concern to the readers, the following changes have been made in the abstract section:

“(..) the mediation analysis showed that inhibition partially mediated the relationship between CRF and language (…) and mathematics (…); similarly, cognitive flexibility mediated CRF’s relationship with language (…) and with mathematics (…); however, a significant relationship remained” (Page 2-3, lines 44-48). 

The last sentence from the abstract is merely speculative and it is not related to the study conclusions.

Authors’ response: Thank you for the comment. We agree regarding the inappropriateness of the sentence. Thus, the sentence has been modified as follows:

“The findings indicated that a significant proportion of the positive influence of CRF on academic achievement was mediated by improvements in inhibition and cognitive flexibility. Thus, improvements in CRF may contribute to increasing academic achievement not only through a direct mechanism but also through improvements in executive functions” (Page 3 lines 52-55).

How was the sampling process conducted? Please provide the inclusion and exclusion criteria.

Authors’ response: Thank you for the thoughtful comment. As suggested, we have provided some detail on the sampling procedure as well as inclusion/exclusion criteria as follows:

Methods

This was a cross-sectional analysis of the baseline data from a cluster-randomized controlled trial (NCT03236337) aimed at assessing the effectiveness of a physical activity program (MOVI-daFit!) to diminish cardiometabolic risk and to improve CRF, executive functioning and academic achievement in schoolchildren. The design, sampling procedures and methods for collecting the variable data have been described elsewhere (Martinez Vizcaíno et al Medicine (Baltimore). 2019 Mar;98(9):e1473) (Page 5, lines 107-108).

Participants 

This study included 570 schoolchildren, aged 8 to 11 years, attending 10 schools in Cuenca, Spain. In these schools, all fourth- and fifth-grade children were invited to participate, except for those who met any of the following exclusion criteria: the children had Spanish learning difficulties; teachers or parents reported that the children had serious physical or mental disorders that could impede their participation in the program activities, or pediatricians reported that the children had chronic disorders such as heart disease, diabetes, or asthma that could prevent their participation in the program activities. Finally, our analysis included 563 participants for whom data on all study variables were collected (Page 5, lines 111-116).

Do the authors have formal assent from the children?

Authors’ response: In this study, only the parents signed a written consent form for their children’s participation. However, every child was asked for their consent to participate before taking each test. Furthermore, children were informed of the characteristics of the study, and their opinion was considered in the design of the intervention. Before starting the MOVI intervention, the members of the research team visited the intervention school to facilitate a consensus among parents, teachers, and children on the types of games and modifications for the playground. During the study, through the data collection process, the children were asked to give us their consent to participate. We thought that this could increase compliance with the program.

We found this issue relevant. Thus, we have added the following sentence in the “Methods” section:

“Every child was verbally asked for consent prior to the completion of each test” (Page 6, lines 123-124).

What is the time gap between academic performance data and CRF and cognitive functions assessments?

Authors’ response: We thank you for your comment. To measure academic achievement, the final grades of the previous year were used. This was a measure that synthesizes students’ work for the whole year. In Spain, a final qualification is given to students as a result of a continuous evaluation for the entire academic year. This measure represents relatively stable behaviour or, in other words, students’ academic background. This grade is given in the month of June and represents students’ work for the whole year. We used that grade to assess academic achievement. We asked schools for this information during the months when the baseline study was performed, from September to October 2017. During these months, data were collected in two sessions two weeks apart. On the first visit, physical tests were performed, including CRF. On the second visit, participants completed a cognitive evaluation.

We agree that this measure of academic achievement could have several limitations. We discuss it below in another comment.

Line 107 Please change VO2max to VO2max

Authors’ response: Done. Thank you.

Line 109: What app was used to assess executive functions? Please change from Ipad to tablet. Describe model and manufacturer in parentheses.

Authors’ response: Thank you. As suggested, we have included a description of the app used for assessing executive functions and replaced Ipad with tablet in the methods section as follows:

“NIH Toolbox (NIH Toolbox in Spanish, v. 1.8)” (page 7, lines 151).

“All measurements were performed (…) using a tablet (iPad Pro, Apple, Inc.) (page 7, lines 152-153)”.

It is mandatory to the authors to provide more information about the measurements of academic performance.

Authors’ response: Thank you for the suggestion. We have modified the text as follows:

“Academic achievement was measured with data provided by the schools. These data consisted of children’s final grades in language and mathematics from the previous year (2016/17, 3rd and 4th grades). In Spain, academic achievement is measured on a 0 to 10 scale, with 10 being the maximum grade. Final grades represent the student´s work in an entire academic year, as assessed by the teacher in a continuous evaluation process” (Page 6-7, lines 139-143).

Mother educational level is one of the main predictor of children academic performance. The authors should used this data as a controlling variable instead of a SES index, otherwise, the authors must present a better rationally for using this index.

Authors’ response: We sincerely appreciate the good suggestion. We have repeated all analyses using mother educational level as a control variable.

Changes can be observed in the methods section (Page 8, 183-191; Page 10, 24’), Table 1 (Page 11); Table 2 (Page 13); Table S1 and S2 (S1 and S2 Table Supplementary material) and Table 3 (Page 18).

Why partial correlations were not adjusted for SES?

Authors’ response: This comment seems appropriate; thus, mother educational level has been included as a covariate in Table 2 (Page 13).

Why the authors opt for using quartiles?

Authors’ response: We appreciate the comment. Although it has been proposed to use cut offs for dichotomizing CRF levels (please see Ruiz JR et al Br J Sports Med. 2016;50:1451-1458), the aim of categorizing variables measured on an interval scale was to test differences between low, middle and high fitness levels. For this, several authors have used quartiles as a standard procedure (e.g., Álvarez-Bueno et al., 2019. Cardiorespiratory fitness as a mediator of the relationship between birth weight and cognition in school children. Psychology research and behaviour management, 12, 255), since there is no universally accepted procedure for this aim.

To clarify the aims of these ANCOVA procedures, which may be unclear, the following sentences were included in the methods section:

“Because there are no universally accepted cut offs for categorizing low, middle and high CRF and executive performance, we used quartiles for this aim as follows: low (Q1), middle (Q1-Q2) and high (Q4). Using these categories, ANCOVA models were estimated to examine differences in academic achievement (language and mathematics) by CRF and executive function performance (inhibition, cognitive flexibility and working memory). These ANCOVA models were also used to examine the potential confounding or mediating role of executive functions in the relationship between CRF and academic achievement” (Page 9, lines 203-209). 

“In these models, language and mathematics grades were used as dependent variables and CRF, inhibition, cognitive flexibility and working memory categories as fixed factors” (Page 9, lines 209-211).

“For the rest of the analyses, continuous variables were used” (Page 9, lines 217-218).

Did the authors calculated a priori sample size? What is the statistical power of the analyses? I am not an expert in statistical, but I am not confident the authors made the right choices.

Authors’ response: The sample size was estimated not for mediation analyses but for a cluster randomized trial belonging to a representative sample from Cuenca, Spain. A new reference has been included in which the study protocol is described in more depth.

However, the precision of the confidence intervals, as well as the recommended sample size for the mediation analysis (Fritz MS, MacKinnon DP. Required sample size to detect the mediated effect. Psychol Sci. 2012; 18(3):233-239), indicate that the sample size is large enough for the analyses included in this manuscript.

The authors should consider to present the data with the level of accuracy from the instrument used to measure the outcomes (e.g: there is no decimal places for age in years).

Authors’ response: Thank you for the comment. As suggested, since age was obtained in months and the cut offs for classifying weight status in children are estimated for age in months (Cole TJ, Lobstein T. Extended international (IOTF) body mass index cut-offs for thinness, overweight and obesity. Pediatr Obes. 2012; 7(4): 284-94), we have included age in months in the tables and analyses (Table 1, Page 10).

I suggest the authors to avoid the VO2 calculation using equation; they can simple use the last complete stage on shuttle run test.

Authors’ response: Thank you for the comment. We assessed both variables before deciding to use VO2max. The main reason for choosing this variable was that VO2max is the most continuous variable, whereas stage is a variable that represents the stage reached. We found this kind of variable more adapted to the procedures we were planning to perform. Another reason to choose VO2max is that this is an indicator that considers weight (mL/kg/min). In a population with a considerable prevalence of overweight/obesity (28%), this is an important fact to consider. However, all analyses were performed with both variables with similar results. We also checked previous research in which this variable was used for analyses (e.g., Aadland et al., 2017; Castro-Piñero et al., 2017; Álvarez-Bueno et al., 2019; Scott et al. 2016).

What is the criteria for SES categorization?

Authors’ response: Following your suggestions, we used mother educational level instead of SES as a control variable in all analyses. The SES variable has been relegated to descriptive purposes in Table 1. For this aim, we used an index developed by the Spanish Society of Epidemiology and the Spanish Society of Family and Community Medicine following criteria proposed by Goldthorpe that are commonly used in UK (please see Domingo‐Salvany A et al. Proposal for a social class measure. Working Group of the Spanish Society of Epidemiology and the Spanish Society of Family and Community Medicine. Aten Primaria. 2000; 25: 350‐ 363). Thus, we have included this information in the manuscript as follows: 

“To asses socioeconomic status, we surveyed parents using the scale proposed by the Spanish Society of Epidemiology [22], with items related to employment and the highest level of education obtained by both the father and mother. Five SES categories were obtained: lower, lower middle, middle, upper middle and upper. Because most authors have reported that mother education level is a strong predictor of children’s academic achievement, we obtained this variable using items related to the highest level of education obtained by the mother from this questionnaire. We thus obtained a five-category index: no reading/no studies, elementary studies, secondary studies, high school and university studies” (Page 8, lines 183-191).

Would be of value if the authors should use some data related to body composition in the analyses (at least a BMI could be presented).

Authors’ response: Thank you for the recommendation. We have included BMI in Table 1 (Page 11) and Table 2 (Pages 13). As the association between BMI and the dependent variables was not significant, we decided not to include this variable as a covariate in the subsequent analyses.

The results section is hard to flow. Firstly, I believe the authors should rethink their statistical procedures. I strongly suggest the authors to avoid categorization and use continuous data.

Authors’ response: Thank you for the comment. According to your suggestion, our analyses incorporated the variables at the interval scale for testing the association between academic achievement, cognitive performance and cardiorespiratory fitness and for estimating the regression coefficients included in the mediation analyses. However, to provide a more comprehensive analysis, we tested differences in the means of the academic achievement variables by executive functions and physical fitness categories, which was our first objective (to analyse whether there are differences in academic achievement in language or mathematics between low- and high-fit children and between low- and high-executive-functioning children). Likewise, we examined the potential confounder or mediator role of executive functions by using ANCOVA models in which cardiorespiratory fitness was categorized as low, middle and high; this is a common practice and can provide readers a comprehensive view and complement interval-scale analyses of changes in the relationship between independent and dependent variables.

This procedure has been performed in previous studies (e.g., Diez Fernandez et al. Diabetes Care. 2014;37(3):855-62).

However, we agree with you that this section could be difficult to follow. Thus, we have replaced the ANCOVA with figures (Fig 1 and Fig 2) in the main text. The table is included as supplementary material (S1 and S2), and the ANCOVA procedures are described in the methods section as follows:

‘Because there are no universally accepted cut offs for categorizing low, middle and high CRF and executive performance, we used quartiles for this aim as follows: low (Q1), middle (Q1-Q2) and high (Q4). Using these categories, ANCOVA models were estimated to examine differences in academic achievement (language and mathematics) by CRF and executive function performance (inhibition, cognitive flexibility and working memory). These ANCOVA models were also used to examine the potential confounding or mediating role of executive functions in the relationship between CRF and academic achievement. In these models, language and mathematics grades were used as dependent variables and weight status, CRF, inhibition, cognitive flexibility and working memory categories as fixed factors. In these models, language and mathematics were used as dependent variables and CRF, inhibition, cognitive flexibility and working memory categories as fixed factors, controlling for age and mother educational level (Model 1). Further adjustment for the executive function index or CRF, depending on the fixed factor, was included in Model 2. Pairwise post hoc hypotheses were tested using the Bonferroni correction for multiple comparisons. Effect size partial eta2 for ANCOVA test was calculated. Indicates small effects values of 0.01, intermediate effects 0.06 and strong effects >0.014 (Cohen J. Statistical power analysis for the behavioral sciences. New York: Academic Press 1988, pp 280-287). These ANCOVA models were estimated for the entire sample and stratified by gender. For the rest of the analyses, continuous variables were used’ (Page 9, lines 203-218).

The correlations may be presented in figures.

Authors’ response: We appreciate your comment. We have added the following correlation matrix plot, performed in Python, to the supplementary material (S1 Fig).

If the authors accept my suggestion and change the statistical approach, the discussion should be rewritten in the light of the new results.

Authors’ response: Thank you for the comment. As suggested, we have added the following sentences to the Discussion section:

“Our study suggested that the association between CRF and academic achievement in mathematics and language was partially mediated by inhibition and cognitive flexibility, even after controlling for age and mother educational level” (Page 19, lines 358).

“Our results indicated that the mediator effect of executive function between CRF and academic achievement was approximately 13.4 to 36 %” (Page 20, lines 396-397).

“However, mediation analyses revealed a stronger mediating role of executive functions in the relationship between fitness and academic achievement in mathematics in boys than in girls after controlling for mother educational level. This might be due to the higher influence of mother educational level on girls than on boys” (Page 21, lines 415-418).

For me the main limitations are: indirect assessment of CRF and academic performance; If academic performance and CRF are not assess in the same time point (as I am assuming), it could also be a strong limitation of the present investigation.

Authors’ response: Your comment is greatly appreciated. We have added the following sentences to “Strengths and limitations” in the Discussion section:

“Another limitation of the study was that both CRF and academic performance were not assessed simultaneously, and they were assessed using indirect procedures, i.e., the 20-meter shuttle run test and teacher evaluations. We should consider that academic achievement is the result of a long-term learning process, and the cross-sectional evaluation actually implies the final outcome of a 1-year learning follow up. Regarding the validity of the 20-meter shuttle run test to estimate CRF, this is the recommended test in the ALPHA battery, since the direct measurement of VO2max in population-based studies is unfeasible” (Page 21, lines 425-431).

Response to Reviewer 3 Comments

Executive functions as a mediator in the relationship between cardiorespiratory fitness and academic achievement in schoolchildren.

Comments

Title: can the title be changed to a declarative statement (e.g. Executive functions mediate the relationship between…) and better specify the study population (add country)?

Authors’ response: We appreciate the suggestion, and the following change has been made:

“Executive functions mediate the relationship between cardiorespiratory fitness and academic achievement in Spanish schoolchildren aged 8 to 11 years” (Page 1-2).

Abstract:

“It has been hypothesized that” or previous researched have actually studied this relationship?

Authors’ response: Thank you for the comment. We have modified the sentence as follows:

“Previous research has studied the influence of physical fitness on academic achievement through executive functions” (Page 2, lines 25-26).

“the associations between cardiorespiratory fitness (CRF) and the executive function domains with academic achievement and…” is unclear. Associations between A and B with C? Do you mean interrelationships (e.g. A and B, A and C, B and C)?

Authors’ response: Thank you for the comment. As suggested, we have modified sentence as follows:

“This study aimed to analyze how cardiorespiratory fitness (CRF) and executive functions are associated with academic achievement” (Page 2, lines 28-29)

Line 26 – Add “This” before cross-sectional.

Authors’ response: Done. Thank you for the comment.

Can you add a little more detail on the measurement of CRF and EF? Just add shuttle-run and NIH Toolbox.

Authors’ response: Thank you for the comment. We now include more information on the measures of CRF and EF in the abstract, which has been modified as follows:

“CRF (20-meter shuttle run test) and executive functions (inhibition, cognitive flexibility and working memory, NIH Toolbox battery in Spanish, v 1.8; iPad Pro, Apple, Inc.) were measured” (Page 2, lines 34-36).

Can you add effect sizes to the abstract?

Authors’ response: Thank you. As suggested, we have included ES information in the abstract, and in table S1 (supplementary material), partial eta squared has been included:

“…children in higher categories of CRF, inhibition, cognitive flexibility and working memory than in children in lower categories. The effect sizes were small to medium (p < 0.05, partial eta squared: from 0.05 to 0.12) (Page 2, lines 43-44). 

And we have added the following sentence (in Data analyses section, page 9, lines 215-216):

“…Effect size partial eta2 for ANCOVA test was calculated. Values of 0.01, 0.06, and 0.14 to indicate small, medium, or large effects (Cohen J. Statistical power analysis for the behavioral sciences. New York: Academic Press 1988, pp 280-287)”.

Line 36 – this sentence errs on the causal side and; would reword to state that children in this study with higher CRF and EF had higher math and language scores. It’s just an association in this sample.

Authors’ response: Thank you for the comment. We have rephrased the sentence to avoid establishing a causal relation, as follows:

“Children in higher categories of both CRF and executive function showed higher grades in mathematics and language” (Page 3, lines 51-52).

Line 38 – mediated (past tense); otherwise it sounds like a population inference

Authors’ response: Thank you for the suggestion. The error has been modified.

Introduction

You may also want to cite

• Scott et al. 2016, Med Sci Sport Exer – Cardiorespiratory fitness is associated with better executive function in young women

• Scott et al. 2016, Journal of Nutrition – Combined iron deficiency and low aerobic fitness doubly burden academic performance among women attending university (see fig 3 for mediation)

Authors’ response: The comment is greatly appreciated. We found that those references support the relationship between CRF and cognition. Thus, we have included the references proposed by the reviewer in the introduction section (Page 4, lines 69, 76) and included references 6 and 11 in the bibliography.

Line 73 – same problem as in abstract: Associations between A and B with C?

Authors’ response: Thank you for the comment. We have reformulated the sentence as follows:

“the association between CRF and academic achievement (in language and mathematics) and the association between the three executive function domains and academic achievement” (Page 5, lines 92-93).

Methods

Line 98 – “in standardized conditions”? Do you mean “following standard procedures”? Was this done by nurses or the researchers?

Authors’ response: Thank you for the comment. We have rephrased the paragraph as follows:

“The research was carried out by a multidisciplinary team composed of nurses, psychologists, teachers and physical activity experts. Each test of the study was carried out by the most appropriate team members (e.g., anthropometric tests were carried out by nurses, cognitive tests by psychologists, physical activity tests by sports science educators) (Page 6, lines 125-128).

20 trials on the flanker task does not seem sufficient. Were multiple blocks performed? If you used only 20 trials, can you cite other studies to support that this is acceptable?

Authors’ response: Thank you for the comment. In our study, to measure cognitive inhibition, we used the NIH Flanker Inhibitory Control and Attention Test. This is a test included in a wider battery of tests aimed at assessing cognitive function in people aged 3-85 years. The validation process has been described elsewhere (Weintraub S, Dikmen SS, Heaton RK, Tulsky DS, Zelazo PD, Bauer PJ, et al. Cognition assessment using the NIH Toolbox. Neurology. 2013;80(11 Suppl 3):S54-64.), and the authors conclude that 20 trials is appropriate. In children aged 3-7, two blocks of trials are performed. The first one consists of a set of 20 fishes, and if a participant scores ≥ 90% on the fish stimuli, 20 additional trials with arrows are presented. Before the 20-trial block, 4 practice trials are performed in all tests. After checking validation and information about the NIH Toolbox, we found that the NIH Toolbox was a reliable measure for cognition.

To clarify this concern, we have added the following references supporting the validity and suitability of this test:

Zelazo PD, Anderson JE, Richler J, Wallner-Allen K, Beaumont JL, Weintraub S. II. NIH Toolbox cognition battery (cb): measuring executive function and attention: NIH Toolbox cognition battery (CB). Monographs of the Society for Research in Child Development. 2013;78(4):16-33

Brito NH, et al. Associations between cortical thickness and neurocognitive skills during childhood vary by family socioeconomic factors. Brain and cognition. 2017, vol. 116, p. 54-62.

Akshoomoff N, Brown T, Bakeman R, Hagler D. Developmental differentiation of executive functions on the NIH Toolbox Cognition Battery. Neuropsychology. 2018. 32(7), 777-7783.

Calderon J, Bellinger DC, Hartigan C, Lord A, Stopp C, Wypij D, Newburger JW. Improving neurodevelopmental outcomes in children with congenital heart disease: protocol for a randomised controlled trial of working memory training. 2019. BMJ open, 9(2).

Would it not be useful to treat accuracy and speed as separate variables? Were children with <80% accuracy dropped from the analysis?

Authors’ response: Thanks for the comment. In order to clarify this concern, we have included the following information about the convenience or not of jointly assessing accuracy and speed in the limitation section:

Because some authors recommend the measurement of accuracy and reaction time as separate variables, the assessment of these cognitive abilities as an unique score could be seen as a limitation; however, the NIH Toolbox´s total score has been used and validated in several studies and allows the avoidance of a ceiling effect in accuracy, which is typically achieved by the most of participants of this age” (Page 22, lines 436-440).

“For children scoring < 80%, a total score considering accuracy was performed” (Page 7, 164-165; 173-174).

Line 132 – what are aliments? Food items?

Authors’ response: Yes, we mean “food items”. We apologize for the confusion, and this text has been corrected. Thank you for the comment.

Please provide more information on the SES index. What were the education and employment categories and how were these used to construct the SES index? You say highest level of education in their family (all family members) and then either father or mother. Which was it? Too bad you didn’t ask about their income.

Authors’ response: Thank you for the thoughtful comment. Because most authors recognize not a whole SES assessment but mother education level as the main predictor of academic achievement, we have used SES categories for the description of the study sample characteristics and mother education level (gathered from the items included in the SES measurement) as control variables. We used an index developed by the Spanish Society of Epidemiology and the Spanish Society of Family and Community Medicine, following criteria proposed by Goldthorpe that are commonly used in the UK (please see Domingo‐Salvany A et al. Proposal for a social class measure. Working Group of the Spanish Society of Epidemiology and the Spanish Society of Family and Community Medicine. Aten Primaria. 2000; 25: 350‐ 363), as used in previous studies.

This information has been included in the methods section as follows:

“To asses socioeconomic status, we surveyed parents using the scale proposed by the Spanish Society of Epidemiology [22], with items related to employment and the highest level of education obtained by both the father and mother. Five SES categories were obtained: lower, lower middle, middle, upper middle and upper. Because most authors have reported that mother education level is a strong predictor of children’s academic achievement, we obtained this variable using items related to the highest level of education obtained by the mother from this questionnaire. We thus obtained a five-category index: no reading/no studies, elementary studies, secondary studies, high school and university studies” (Page 8, 183-191).

The model descriptions are unclear to me. All of your variables are continuous, so why not just use OLS regressions with multiple predictors? “We estimated ANCOVA models to test mean differences in language and mathematics by categories of CRF and executive function domain, controlling for age, pubertal status and SES (Model 1); further adjustment for the executive function index or CRF, depending on the fixed factor, was included in Model 2.”

Authors’ response: Your concern is valid. To address it, we have included a paragraph in the introduction section justifying the mediation analysis.

“Health studies commonly control for confounding or mediator variables using multivariate methods such as multiple linear regression, logistic regression, or ANCOVA, depending on the study aims and/or whether the dependent variable is quantitative or categorical. Mediation analysis is a statistical procedure suitable for clarifying whether a third variable underlies the relationship between two variables and determining the extent to which this relationship can be modified, mediated, or confounded by this third variable (Baron RM, Kenny DA. The moderator-mediator variable distinction in social psychological research: conceptual, strategic, and statistical considerations. J Pers Soc Psychol 1986;51:1173–1182). We consider that a mediation effect occurs when a significant proportion of the influence of the independent variable on the dependent variable takes place through a third variable (mediator)” (Page 5 95-102). 

Does adjusting for pubertal status (I think you mean stage; status implies a dummy variable, i.e. pre/post pubertal) really add much? You already adjust for age, which I would assume is colinear with pubertal status, though I could be wrong.

Authors’ response: We apologize for the mistake, which has been corrected. We mean stage, not “status”.

According to your suggestion, we have repeated some analyses, and we find reasons to remove pubertal stage from the adjustment. First, most children in the sample had very low pubertal stage development (86% presented a pubertal stage <=2); moreover, this variable was closely associated with age. Thus, as the reviewer recommends, pubertal stage has been removed as a control variable. These changes can be observed in the methods section (Page 8, lines 194-196), Table 2 (Page 13), Table 3 (Page 18) and Table S1 and S2 (Supplementary material).

Results

Table 1

-add BMI

Authors’ response: Done. BMI is now included in Table 1 (Page 11) and Table 2 (Page 13). Thanks for the comment.

-add % falling into fitness categories based on CRF? VO2max of 46 seems fairly fit, but I don’t know what the ACSM cutoffs are in this age group. Need to get a sense of whether sample is fit or unfit.

Authors’ response: Thanks for the comment. 

CRF categories were made attending to quartiles in our sample because at this age, there are no ACSM categories. Some authors have proposed points that could indicate a “red flag” of cardiovascular disease risk (e.g., Ruiz et al., 2015).

As suggested, to provide the readers an estimation of the children’s fitness levels, we have included the VO2max levels corresponding to each category in Table 1 (Page 11).

-need units for EF indicators. I assume accuracy is % incorrect responses? This is typically expressed as proportion correct.

Authors’ response: Thank you for the comment. Accuracy refers to total correct responses. Reaction time is time in milliseconds. The total score is a score that takes into account accuracy and time reaction, calculated by the app using the previously explained algorithm. It provides a score from 0-10.

To improve understandability, we have included the appropriate units of EF indicators in Table 1 (Page 11).

Table 3 is very hard to follow. I suggest moving this to supplement and creating a figure or figures for the main paper.

Authors’ response: Thank you for the comment. Table 3 has been moved to the supplementary material (S1 and S2 Tables), and a figure synthesizing the differences in academic achievement between children in lower and higher categories of CRF and EF has been included (Fig 1 and Fig 2).

Discussion – can you comment on what may mediate the other 70-85% of the relationship between CRF and language/math that is not explained by executive function, age, pubertal status, and SES?

Authors’ response: Thank you for the recommendation. As suggested, we have included some sentences addressing this concern in the discussion section as follows:

 “Our results indicated that the moderator effect of EF between CRF and academic achievement was approximately 13.4 to 36 %. Several other fitness-related factors could influence academic achievement, such as motor skills. Two studies showed that motor skills, rather than aerobic fitness or physical activity, had the strongest association with executive functions and academic performance [33,34]. A review study [35] proposed that both CRF and motor skills were related to academic performance and that the influence of both could affect cognition and academic performance in different ways. Moreover, a mediation study showed that motor coordination—not cardiovascular endurance or muscular strength—was the only motor ability that accounted for a significant percentage of the variance in academic achievement [36]. Other variables that have been demonstrated to be related to academic achievement, possibly influencing the relationship between CRF and academic achievement, are physical activity and general health improved by cardiorespiratory fitness (Marques, Santos, Hillman & Sardinha, 2018)” (Page 19, lines 396-406).

---

## [Decision Letter · Decision Letter 1]

7 Feb 2020

PONE-D-19-14578R1

Executive functions mediate the relationship between cardiorespiratory fitness and academic achievement in Spanish schoolchildren aged 8 to 11 years.

PLOS ONE

Dear Mrs. SANCHEZ,

Thank you for submitting your manuscript to PLOS ONE. After careful consideration, we feel that it has merit but does not fully meet PLOS ONE’s publication criteria as it currently stands. Therefore, we invite you to submit a revised version of the manuscript that addresses the points raised during the review process.

We would appreciate receiving your revised manuscript by Mar 23 2020 11:59PM. To enhance the reproducibility of your results, we recommend that if applicable you deposit your laboratory protocols in protocols.io, where a protocol can be assigned its own identifier (DOI) such that it can be cited independently in the future. For instructions see: http://journals.plos.org/plosone/s/submission-guidelines#loc-laboratory-protocols

We look forward to receiving your revised manuscript.

Kind regards,

Tarek K Rajji

Academic Editor

PLOS ONE

Reviewers' comments:

Reviewer's Responses to Questions

**Comments to the Author**

1. If the authors have adequately addressed your comments raised in a previous round of review and you feel that this manuscript is now acceptable for publication, you may indicate that here to bypass the “Comments to the Author” section, enter your conflict of interest statement in the “Confidential to Editor” section, and submit your "Accept" recommendation.

Reviewer #4: (No Response)

Reviewer #5: (No Response)

2. Is the manuscript technically sound, and do the data support the conclusions?

Reviewer #4: Yes

Reviewer #5: Partly

3. Has the statistical analysis been performed appropriately and rigorously? 

Reviewer #4: Yes

Reviewer #5: No

4. Have the authors made all data underlying the findings in their manuscript fully available?

Reviewer #4: Yes

Reviewer #5: Yes

5. Is the manuscript presented in an intelligible fashion and written in standard English?

Reviewer #4: Yes

Reviewer #5: Yes

6. Review Comments to the Author

Reviewer #4: Important note: This review pertains only to ‘statistical aspects’ of the study and so ‘clinical aspects’ [like medical importance, relevance of the study, ‘clinical significance and implication(s)’ of the whole study, etc.] are to be evaluated [should be assessed] separately/independently.

Please note that it is preferable [refer to item 1b of CONSORT checklist 2010: Structured summary of trial design, methods, results, and conclusions; though article type is ‘Research Article’] to divide the ABSTRACT with small sections like ‘Objective(s)’, ‘Methods’, ‘Results’, ‘Conclusions’, etc. which is the accepted practice of most good/standard journals [including PLOS]. Your ABSTRACT is well drafted but assay type.

This being a cross-sectional study, there are few ‘design’ issues involved, regarding which, I am sure, authors are well aware [clear as it is mentioned in ‘Strengths and limitations’ section later]. All sorts of biases [more likely ‘selection’ biases] are common occurrence in such cross-sectional studies. A cross-sectional study is easy to conduct but generally is of very limited value [since in no way any cross-sectional study will yield confirmatory conclusion(s) mainly because of absence of temporal sequence]. Since any cross-sectional study is considered as a low level of [scientific] evidence, such design is used only if that is what is possible [that is the only possibility]. {This point is raised here particularly in the backdrop of certainty needed in finding answer(s) to posed question(s)}.

Is the reference number 18 [Baron RM, Kenny DA. The Moderator-Mediator Variable Distinction in Social Psychological Research: Conceptual, Strategic, and Statistical Considerations] complete? (this question is raised because this reference is of vital importance, in my opinion). Although this study is off-shoot of another study [MOVI-daFIT! Intervention: Rationale and design of a cluster randomized controlled trial testing the effects on improving adiposity, cognition, and subclinical atherosclerosis by increasing cardiorespiratory fitness in children], it seems to be methodologically well conducted/executed one. My only query is : why line numbers are not given for ‘discussion’ section?

Reviewer #5: This manuscript presents baseline data from a randomized controlled trial and examined the mediating role of executive functioning in the relationship between cardiovascular fitness and academic achievement. This study is well powered and answers an interesting research question. Further thoughts are presented below:

Major Issues:

There is little to no background given on why males and females are examined separately. This should be expanded on in the introduction since it is such a large piece of the analytic strategy.

The results and data analysis are extremely unclear. Was a multiple mediation analysis conducted or were separate mediation analyses conducted for each potential mediator and each DV? It appears like separate mediation analyses were conducted. It would seem like a better analytic approach would be to examine a multiple mediation model and determine which EF domains are mediating the relationship instead of looking at them all separately. The data analysis section should be written more clearly and I would suggest reconsidering the analytic plan.

Minor Issues:

Line 55 – 60: There are many more cognitive processes than the 3 described. Even when considering EF there are many more components to EF than just the 3 that are described.

Line 69: “Indirect effect of executive functions” is not fully accurate. Indirect effect is really the effect of IV through the pathway of mediator

Line 85-89: Discussion of mediation analysis seems unnecessary in the intro and this discussion is somewhat misleading. Mediation cannot determine whether a variable is a true mediator or a confounding variable

Line 180 – strong effects should read 0.14

208 – “There were no significant differences…” should be rephrased given that there were several differences between gender

7. PLOS authors have the option to publish the peer review history of their article (what does this mean?). If published, this will include your full peer review and any attached files.

Reviewer #4: Yes: Dr. Sanjeev Sarmukaddam

Reviewer #5: No

---

## [Author Response · Author response to Decision Letter 1]

26 Feb 2020

Response to reviewer 4 Comments

Important note: This review pertains only to ‘statistical aspects’ of the study and so ‘clinical aspects’ [like medical importance, relevance of the study, ‘clinical significance and implication(s)’ of the whole study, etc.] are to be evaluated [should be assessed] separately/independently.

Please note that it is preferable [refer to item 1b of CONSORT checklist 2010: Structured summary of trial design, methods, results, and conclusions; though article type is ‘Research Article’] to divide the ABSTRACT with small sections like ‘Objective(s)’, ‘Methods’, ‘Results’, ‘Conclusions’, etc. which is the accepted practice of most good/standard journals [including PLOS]. Your ABSTRACT is well drafted but assay type.

Authors response: We sincerely appreciate the reviewer´s comment; thus, as suggested, we have structured the abstract (see Page 2, Abstract section).

This being a cross-sectional study, there are few ‘design’ issues involved, regarding which, I am sure, authors are well aware [clear as it is mentioned in ‘Strengths and limitations’ section later]. All sorts of biases [more likely ‘selection’ biases] are common occurrence in such cross-sectional studies. A cross-sectional study is easy to conduct but generally is of very limited value [since in no way any cross-sectional study will yield confirmatory conclusion(s) mainly because of absence of temporal sequence]. Since any cross-sectional study is considered as a low level of [scientific] evidence, such design is used only if that is what is possible [that is the only possibility]. {This point is raised here particularly in the backdrop of certainty needed in finding answer(s) to posed question(s)}.

Authors response: Thank you for the comment. Although temporal ambiguity is certainly a core limitation to stablish causality in the relationship analyses, our mediation models, as some authors such as Stillman et al. 1, support the usefulness of these statistical mediation models to test plausibility of causal models, when experimental manipulation (the gold standard for assessing causality) is not available. Moreover, although this temporal limitation cannot be ruled out, it seems more plausible, according to the existing literature 2–4, that higher cardiorespiratory fitness causes better academic achievement, and not the opposite.

Therefore, as suggested, we have rephrased some sentences related to the reviewer´s concern:

“Thus, this study supports the hypothesis that improvements in CRF may contribute to increasing academic achievement not only through a direct mechanism but also through improvements in executive functions” (Page 3, line 44).

 “Although previous studies support a directional relationship between physical activity and fitness to academic achievement, the cross-sectional nature of our analyses prevented us from making cause–effect inferences, since temporal ambiguity bias cannot be completely ruled out” (Page 17, lines 369-372).

1. Stillman, C. M., Cohen, J., Lehman, M. E. & Erickson, K. I. Mediators of Physical Activity on Neurocognitive Function: A Review at Multiple Levels of Analysis. Frontiers in Human Neuroscience 10, (2016).

2. Álvarez-Bueno, C. et al. The Effect of Physical Activity Interventions on Children’s Cognition and Metacognition: A Systematic Review and Meta-Analysis. Journal of the American Academy of Child & Adolescent Psychiatry 56, 729–738 (2017).

3. Donnelly, J. E. et al. Physical Activity, Fitness, Cognitive Function, and Academic Achievement in Children: A Systematic Review. Med Sci Sports Exerc 48, 1197–1222 (2016).

4. Hillman, C. H. et al. Effects of the FITKids randomized controlled trial on executive control and brain function. Pediatrics 134, e1063-1071 (2014).

Is the reference number 18 [Baron RM, Kenny DA. The Moderator-Mediator Variable Distinction in Social Psychological Research: Conceptual, Strategic, and Statistical Considerations] complete? (this question is raised because this reference is of vital importance, in my opinion). Although this study is off-shoot of another study [MOVI-daFIT! Intervention: Rationale and design of a cluster randomized controlled trial testing the effects on improving adiposity, cognition, and subclinical atherosclerosis by increasing cardiorespiratory fitness in children], it seems to be methodologically well conducted/executed one. My only query is: why line numbers are not given for ‘discussion’ section?

Authors response: Thank you, we apologize for the inconvenience. Both typos have been corrected (please, see pages 15-19, in discussion section and this reference corrected now reference 37, page 24, lines 521-524). 

37. Baron, R. M. & Kenny, D. A. The Moderator-Mediator Variable Distinction in Social Psychological Research: Conceptual, Strategic, and Statistical Considerations. search. Journal of Personality and Social Psychology 51 1173-1182 (1986).

Response to reviewer 4 Comments

This manuscript presents baseline data from a randomized controlled trial and examined the mediating role of executive functioning in the relationship between cardiovascular fitness and academic achievement. This study is well powered and answers an interesting research question. Further thoughts are presented below:

Major Issues:

There is little to no background given on why males and females are examined separately. This should be expanded on in the introduction since it is such a large piece of the analytic strategy.

Authors response: Thank you. The reviewer’s comments and, in general, his/her involvement in the review process are highly appreciated. In order to duly justify the need to present the results separated by gender, we have modified the paragraph as follows:

“Additionally, no study has provided results separately by gender, even though gender differences in the association between CRF and academic achievement have been reported [17]. Moreover, differential brain maturation and executive function development between boys and girls have been reported [18]” (Page 5, lines 77-80).

The results and data analysis are extremely unclear. Was a multiple mediation analysis conducted or were separate mediation analyses conducted for each potential mediator and each DV? It appears like separate mediation analyses were conducted. It would seem like a better analytic approach would be to examine a multiple mediation model and determine which EF domains are mediating the relationship instead of looking at them all separately. The data analysis section should be written more clearly, and I would suggest reconsidering the analytic plan.

Authors response: The reviewer´s comment seems thoughtful. Thus, we have repeated the analyses according to her/his recommendation. Both models led us to the same conclusions:

A.

B.

Fig 3. Parallel serial mediation model. Cardiorespiratory fitness (independent variable) on academic achievement (dependent variable), through executive functions (mediator variable). Path a, association between independent variable and mediators; path b, association between mediator and dependent variables; path c, total effect between independent and dependent variables; path c´, direct effect of independent variable on dependent variable.

Finally, we have decided to show separate mediation models for each potential mediator in order to provide the reader a more clearly and simply information. Moreover, we have included also parallel serial mediation (as supplementary material) in order to provide the reader, the complete analyses, although statistical significance disappeared by gender, probably due to the lack of statistical power. Thus, we have included some sentences regarding this concern:

“Simple mediation models were tested, by introducing independently the three different mediating variables. Then, a parallel serial mediation model was tested in which the three domains of executive function (inhibition, cognitive flexibility and working memory) sere simultaneously entered in the model in such a way that each mediator was controlled for the others” (Page 9, lines 206-210). 

“The mediation model scheme is displayed in Fig 3, and the results of the mediation analysis are shown in Table 3. Inhibition and cognitive flexibility mediated the association between CRF and academic achievement, controlling for age and mother educational level. In the first regression step (path a), CRF was positively related to the executive function domains (p < 0.05). In the second step (path c), the regression coefficients of CRF were positively associated with academic achievement in all mediation models (p < 0.001). In the last regression model (path b), the executive function domains were positively related to the dependent variable (p < 0.001), and the relationship between CRF and academic achievement became weaker (although significant) when the executive function domains were included in the model (path c’). Finally, the indirect effects were statistically significant for inhibition and cognitive flexibility, confirming that executive function domains acted as a partial mediator in these models, except for working memory. The mediation analyses by gender showed that, after controlling for mother educational level, cognitive flexibility was not significant in girls and inhibition in boys. When parallel serial mediation was performed, the results were similar (S2, Fig 4).” (Page 13, line 270-282).

Minor Issues:

Line 55 – 60: There are many more cognitive processes than the 3 described. Even when considering EF there are many more components to EF than just the 3 that are described.

Authors response: Thank you for the comment. The reviewer´s is right. Thus, we have rephrased the sentence as follows:

“Executive functions are a set of cognitive operations, three of the most studied are inhibition, working memory and cognitive flexibility” (Page 4, line 58-59).

Line 69: “Indirect effect of executive functions” is not fully accurate. Indirect effect is really the effect of IV through the pathway of mediator

Authors response: The reviewer´s is right. We have modified the sentence as follows:

“These studies have shown a significant positive relationship between physical fitness and academic achievement mediated by executive functions” (Page 4, line 70-72)

Line 85-89: Discussion of mediation analysis seems unnecessary in the intro and this discussion is somewhat misleading. Mediation cannot determine whether a variable is a true mediator or a confounding variable

Authors response: We appreciate this comment, thus, we have removed that paragraph.

Conversely, we have added a brief commentary about the convenience of this analysis in the discussion section:

 “Mediation analysis has been proposed as an statistical procedure suitable for clarifying whether a third variable underlies the relationship between two variables. Thus, this procedure allows us to consider the significant proportion of the influence of the independent variable (CRF) on the dependent variable (academic achievement) takes place through executive functions (mediator) [37]” (Page 17, line 365-369).

Line 180 – strong effects should read 0.14

Authors response: Thank you for the comment. We apologize for the inconvenience. It has been corrected (Page 9, line 193).

208 – “There were no significant differences…” should be rephrased given that there were several differences between gender

Authors response: Thank you for the comment. We have rephrased the sentence related to the reviewer’s concern:

“There were several differences between gender in (…)” (Page 10, line 216-217)

---

## [Decision Letter · Decision Letter 2]

11 Mar 2020

PONE-D-19-14578R2

Executive functions mediate the relationship between cardiorespiratory fitness and academic achievement in Spanish schoolchildren aged 8 to 11 years.

PLOS ONE

Dear Mrs. SANCHEZ,

Thank you for submitting your manuscript to PLOS ONE. After careful consideration, we feel that it has merit but does not fully meet PLOS ONE’s publication criteria as it currently stands. Therefore, we invite you to submit a revised version of the manuscript that addresses the points raised during the review process.

ACADEMIC EDITOR: Please insert comments here and delete this placeholder text when finished. Be sure to:

Indicate which changes are required versus recommended for acceptanceAddress any conflicts between the reviewsProvide specific feedback from your evaluation of the manuscript

We would appreciate receiving your revised manuscript by Apr 25 2020 11:59PM. To enhance the reproducibility of your results, we recommend that if applicable you deposit your laboratory protocols in protocols.io, where a protocol can be assigned its own identifier (DOI) such that it can be cited independently in the future. For instructions see: http://journals.plos.org/plosone/s/submission-guidelines#loc-laboratory-protocols

We look forward to receiving your revised manuscript.

Kind regards,

Joshua Jones

Academic Editor

PLOS ONE

Journal Requirements:

Additional Editor Comments (if provided):

Reviewers' comments:

Reviewer's Responses to Questions

**Comments to the Author**

1. If the authors have adequately addressed your comments raised in a previous round of review and you feel that this manuscript is now acceptable for publication, you may indicate that here to bypass the “Comments to the Author” section, enter your conflict of interest statement in the “Confidential to Editor” section, and submit your "Accept" recommendation.

Reviewer #4: All comments have been addressed

Reviewer #5: All comments have been addressed

2. Is the manuscript technically sound, and do the data support the conclusions?

Reviewer #4: Yes

Reviewer #5: (No Response)

3. Has the statistical analysis been performed appropriately and rigorously? 

Reviewer #4: Yes

Reviewer #5: (No Response)

4. Have the authors made all data underlying the findings in their manuscript fully available?

Reviewer #4: Yes

Reviewer #5: (No Response)

5. Is the manuscript presented in an intelligible fashion and written in standard English?

Reviewer #4: Yes

Reviewer #5: (No Response)

6. Review Comments to the Author

Reviewer #4: Although I recommend to accept the article for publication in this prestigious journal, I request you to note the following [regarding the article quoted in response to my earlier comment saying that “as some authors such as Stillman et al. support the usefulness of these statistical mediation models to test plausibility of causal models, when experimental manipulation (the gold standard for assessing causality) is not available.”]:

Refer to page 6 of

Stillman, C. M., Cohen, J., Lehman, M. E. & Erickson, K. I. Mediators of Physical Activity on Neurocognitive Function: A Review at Multiple Levels of Analysis. Frontiers in Human Neuroscience 10, (2016) which gives the following account:

Cross-sectional studies utilizing statistical mediation provide a theoretical and mechanistic foundation about the relationships between cardiorespiratory fitness, brain, and cognition. However, their correlational nature leaves open the possibility that the observed behavioral and structural fitness-related differences between high and low-fit groups are caused by some unmeasured factor. RCTs are necessary to account for potential selection bias, as well as to establish a direct, causal relationship in humans between aerobic fitness, brain structure, and cognitive functioning.

Reviewer #5: (No Response)

7. PLOS authors have the option to publish the peer review history of their article (what does this mean?). If published, this will include your full peer review and any attached files.

Reviewer #4: No

Reviewer #5: No

---

## [Author Response · Author response to Decision Letter 2]

13 Mar 2020

Response to reviewer 4: 

Although I recommend to accept the article for publication in this prestigious journal, I request you to note the following [regarding the article quoted in response to my earlier comment saying that “as some authors such as Stillman et al. support the usefulness of these statistical mediation models to test plausibility of causal models, when experimental manipulation (the gold standard for assessing causality) is not available.”]:

Refer to page 6 of

Stillman, C. M., Cohen, J., Lehman, M. E. & Erickson, K. I. Mediators of Physical Activity on Neurocognitive Function: A Review at Multiple Levels of Analysis. Frontiers in Human Neuroscience 10, (2016) which gives the following account:

Cross-sectional studies utilizing statistical mediation provide a theoretical and mechanistic foundation about the relationships between cardiorespiratory fitness, brain, and cognition. However, their correlational nature leaves open the possibility that the observed behavioral and structural fitness-related differences between high and low-fit groups are caused by some unmeasured factor. RCTs are necessary to account for potential selection bias, as well as to establish a direct, causal relationship in humans between aerobic fitness, brain structure, and cognitive functioning.

Authors response: We appreciate the reviewer´s comment. We have considered the difference between what he or she suggested and the idea we were trying to expose. Thus, we have rephrased the following sentences:

“(…) however, cross-sectional studies utilizing statistical mediation provide a theoretical and mechanistic foundation about the relationships between cardiorespiratory fitness, brain, and cognition. Nevertheless, their correlational nature leaves open the possibility that the observed behavioural and structural fitness-related differences between high and low-fit groups are caused by some unmeasured factor” (lines 361-365)

“(…) Randomized controlled trials are necessary to account for potential selection bias, as well as to establish a direct, causal relationship in humans between aerobic fitness, brain structure, and cognitive functioning” (lines 368-370, page).

---

## [Editor Report · Decision Letter 3]

20 Mar 2020

Executive functions mediate the relationship between cardiorespiratory fitness and academic achievement in Spanish schoolchildren aged 8 to 11 years.

PONE-D-19-14578R3

Dear Dr. SANCHEZ,

We are pleased to inform you that your manuscript has been judged scientifically suitable for publication and will be formally accepted for publication once it complies with all outstanding technical requirements.

With kind regards,

Tarek K Rajji

Academic Editor

PLOS ONE
---

## [Editor Report · Acceptance letter]

24 Mar 2020

PONE-D-19-14578R3 

Executive functions mediate the relationship between cardiorespiratory fitness and academic achievement in Spanish schoolchildren aged 8 to 11 years. 

Dear Dr. Sánchez-López:

I am pleased to inform you that your manuscript has been deemed suitable for publication in PLOS ONE. Congratulations! Your manuscript is now with our production department. 

With kind regards,

on behalf of

Dr. Tarek K Rajji 

Academic Editor

PLOS ONE